# *SafeMVDrive*: MULTI-VIEW SAFETY-CRITICAL DRIVING VIDEO GENERATION IN THE REAL-WORLD DOMAIN

## ABSTRACT

Safety-critical scenarios are essential for evaluating autonomous driving (AD) systems, yet they are rare in practice. Existing generators produce trajectories, simulations, or single-view videos—but they don't meet what modern AD systems actually consume: realistic multi-view video. We present SafeMVDrive, the first framework for generating multi-view safety-critical driving videos in the real-world domain. SafeMVDrive couples a safety-critical trajectory engine with a diffusion-based multi-view video generator through three design choices. First, we pick the right adversary: a GRPO-fine-tuned vision-language model (VLM) that understands multi-camera context and selects vehicles most likely to induce hazards. Second, we generate the right motion: a two-stage trajectory process that (i) produces collisions, then (ii) transforms them into natural evasion trajectories—preserving risk while staying within what current video generators can faithfully render. Third, we synthesize the right data: a diffusion model that turns these trajectories into multi-view videos suitable for end-to-end planners. On a strong end-to-end planner, our videos substantially increase collision rate, exposing brittle behavior and providing targeted stress tests for planning modules. Our code and video examples are available at: https://iclr-1.github.io/SMD/.

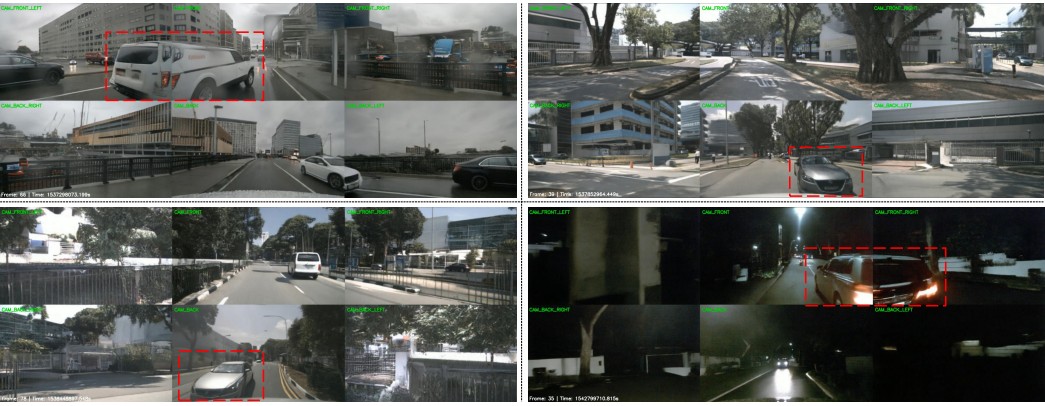

Figure 1: Keyframes from videos generated by **SafeMVDrive**. Red boxes mark safety-critical vehicles: cut-ins (top-left), rapid rear approaches (top-right, bottom-left), and sudden braking (bottom-right). SafeMVDrive generates high-quality multi-view safety-critical videos, providing a critical tool for developing and stress-testing autonomous driving systems. [1]

## 1 INTRODUCTION

Vision-based end-to-end (E2E) autonomous driving (AD) systems map visual inputs directly to driving decisions. They are rapidly advancing and beginning to see deployment in the real world (Zheng et al., 2024a; Hu et al., 2023; Li et al., 2024a; Liao et al., 2025; Jiang et al., 2024). But making

---

[1] Video examples are available at: https://iclr-1.github.io/SMD/#Video Gallery.

them safe remains difficult. What they most need—large-scale, diverse, safety-critical data—is nearly impossible to collect at scale: rare events are rare for a reason, and capturing them in the wild is both costly and dangerous.

Synthetic generation offers a scalable alternative, but existing approaches fall short. Prior work has mostly focused on producing adversarial trajectories with diffusion models (Xu et al., 2025; Zhong et al., 2023b;a; Chang et al., 2024), which can stress-test standalone AD planners but are not directly applicable to E2E AD systems that require video, not trajectories. Simulator-based video generation (Zhang et al., 2024; Abeysirigoonawardena et al., 2019) avoids this mismatch but suffers from a large domain gap between simulation and reality. Deep video generators like Open-Sora (Zheng et al., 2024c) can render accident videos in the real-world domain, yet the results are single-view, low-quality, and unsuitable for multi-view E2E pipelines (Li et al., 2025).

We introduce *SafeMVDrive*, the first framework to generate multi-view safety-critical driving videos in the real-world domain. A straightforward way would be to generate adversarial trajectories and then convert them into control signals for existing multi-view video generators (Wen et al., 2024; Sima et al., 2024; Chen et al., 2024b). This, however, fails for two reasons. First, conventional trajectory generators ignore visual context, relying only on non-visual data like vehicle kinematics and maps, and often select adversarial vehicles that are physically infeasible (see Section 3.2). We instead assist trajectory generation with the visual scene information, using a VLM-based selector to identify the critical vehicle. Second, existing video generators (Chen et al., 2024b; Gao et al., 2024a; Wen et al., 2024) cannot realistically render collisions due to the scarcity of multi-view collision data (Feng et al., 2025). To address this, we design a two-stage trajectory generator: it first produces collisions, then refines them into natural evasions – preserving safety-critical characteristics without requiring explicit crash rendering. Combined with a state-of-the-art multi-view video generator, SafeMVDrive produces high-quality, safety-critical videos in the real-world domain (Figure 1).

The main contributions of our work are summarized as follows:

- In this paper, we introduce **SafeMVDrive**, the first framework capable of generating multi-view safety-critical videos in the real-world domain. The key idea is to couple a safety-critical trajectory simulator with a multi-view driving video generator, and to solve the integration bottlenecks with a VLM-based adversarial vehicle selector and a two-stage evasion trajectory generator.

- We incorporate visual context into the selection of safety-critical vehicles through a fine-tuned vision–language model (VLM). The model is adapted for multi-view driving scene understanding using supervision from an automated annotation pipeline that pairs scenes with their corresponding safety-critical vehicles. This adaptation allows the VLM to reliably identify adversarial vehicles most likely to induce safety-critical scenarios.

- We propose a two-stage trajectory generator that produces collision–evasion trajectories within the capability of existing multi-view video generation models. In stage one, a collision trajectory is generated; in stage two, it is refined into a natural evasion trajectory—preserving safety-critical characteristics while remaining compatible with video generation models.

SafeMVDrive produces high-quality multi-view safety-critical videos in the real-world domain. On the representative E2E autonomous planner UniAD (Hu et al., 2023), our videos induce 30% more collisions than those from original NuScenes data (Caesar et al., 2020), exposing brittle planner behavior. Despite their adversarial nature, the generated videos remain realistic: in user studies, they are rated as natural as videos produced by state-of-the-art models on original benign trajectories.

## 2 RELATED WORK

Safety-critical data generation is essential to enhance end-to-end AD systems' robustness in the real world. The existing work can be categorized into trajectory-based and video-based approaches in terms of their output formats. Trajectory-based approaches generate adversarial trajectories (non-visual), while video-based approaches produce safety-critical driving videos. We note that safety-critical scenarios in autonomous driving are diverse and difficult to enumerate exhaustively. In this paper, the term safety-critical vehicle specifically refers to one scenario type: a safety-critical

situation arising from the interaction between an adversarial vehicle and the ego vehicle due to their trajectory relationship.

**Safety-critical Trajectory generation**: Given an initial traffic context, trajectory-based approaches typically first select an adversarial vehicle and then optimize trajectories that lead to safety-critical situations. Recent works often leverage diffusion models, which excel at capturing highly complex and multimodal trajectory distributions. At test time, loss-gradient guidance (Janner et al., 2022; Zhong et al., 2023b) can be introduced to enable controllable trajectory generation, steering sampled trajectories toward the trajectory data manifold while simultaneously minimizing task-specific loss functions. Methods such as Safe-Sim (Chang et al., 2024) and CTG++ (Zhong et al., 2023a) further incorporate collision-related losses to generate safety-critical trajectories. While these methods show progress, they are incompatible with E2E AD systems, which require video input. Besides, they rely on non-visual data (e.g., vehicle kinematics, maps) to select adversarial vehicles, which often neglects critical visual cues and may result in physically infeasible selections (shown in Figure 3).

**Safety-critical Video generation**: Another research direction is to directly generate safety-critical video data. Some works (Zhang et al., 2024; Xu et al., 2025; Wang et al., 2024) use simulators like Carla (Dosovitskiy et al., 2017) to generate adversarial driving videos. However, these methods' effectiveness is limited by the domain gap between simulation and reality. To generate realistic videos, ADV2 (Li et al., 2025) employs generative models like open-sora (Zheng et al., 2024c), finetuned on real traffic accident data with text captions, to produce adversarial videos from user prompts. However, the videos are low-quality and single-view, limiting their use in E2E AD systems that require high-quality, multi-view inputs.

Unlike existing works, our framework can generate multi-view safety-critical videos in the real-world domain that are compatible with E2E AD systems. This is enabled by strategically integrating a safety-critical trajectory simulator with a multi-view driving video generator, and overcoming key integration challenges through a VLM-based adversarial vehicle selector and a two-stage evasion trajectory generator.

## 3 METHODS

### 3.1 OVERVIEW

Figure 2 shows our framework for generating multi-view safety-critical videos, comprising three parts: (1) a VLM-based adversarial vehicle selector; (2) a two-stage evasion trajectory generator; and (3) a trajectory-to-video generator. The input is single-frame holistic information of an initial scene, combining visual data (multi-view images) and non-visual data (camera parameters, vehicle states, and road maps)—all available in datasets like NuScenes (Caesar et al., 2020), Waymo (Sun et al., 2020), and Argoverse2 (Wilson et al., 2023). First, we mark vehicles within distance $D$ from the ego vehicle with ID-labeled 2D boxes in the multi-view images. The images are then fed into the VLM-based selector to identify the adversarial vehicle $V_{adv}$. With $V_{adv}$'s ID, the two-stage evasion trajectory generator can produce safety-critical trajectories: the first stage generates a collision trajectory where $V_{adv}$ collides with the ego vehicle $V_{ego}$; the second converts it into a realistic evasion trajectory using our proposed method. The generated trajectories are then converted to control signals that guide a diffusion-based video generator to synthesize realistic safety-critical multi-view videos.

### 3.2 VLM-BASED ADVERSARIAL VEHICLE SELECTOR

The first step in generating safety-critical data is selecting the adversarial vehicle $V_{adv}$ from the initial scene. Prior methods rely on simple heuristic methods using non-visual data like vehicle kinematics and maps, such as choosing the nearest neighbor vehicle, applying fixed distance-velocity rules (Zhong et al., 2023a), or selecting a random nearby lane vehicle (Chang et al., 2024). However, these heuristic methods lack crucial visual cues and fail to capture complex driving scenarios, often resulting in physically infeasible selections (shown in Figure 3).

To address the aforementioned problems, we propose incorporating visual information into adversarial vehicle selection by leveraging the scene understanding capabilities of Vision-Language Models (VLMs) (Chen et al., 2024a). Specifically, we introduce a VLM-based selector that selects the critical adversarial vehicle using the multi-view images in the initial scene. To aid comprehension

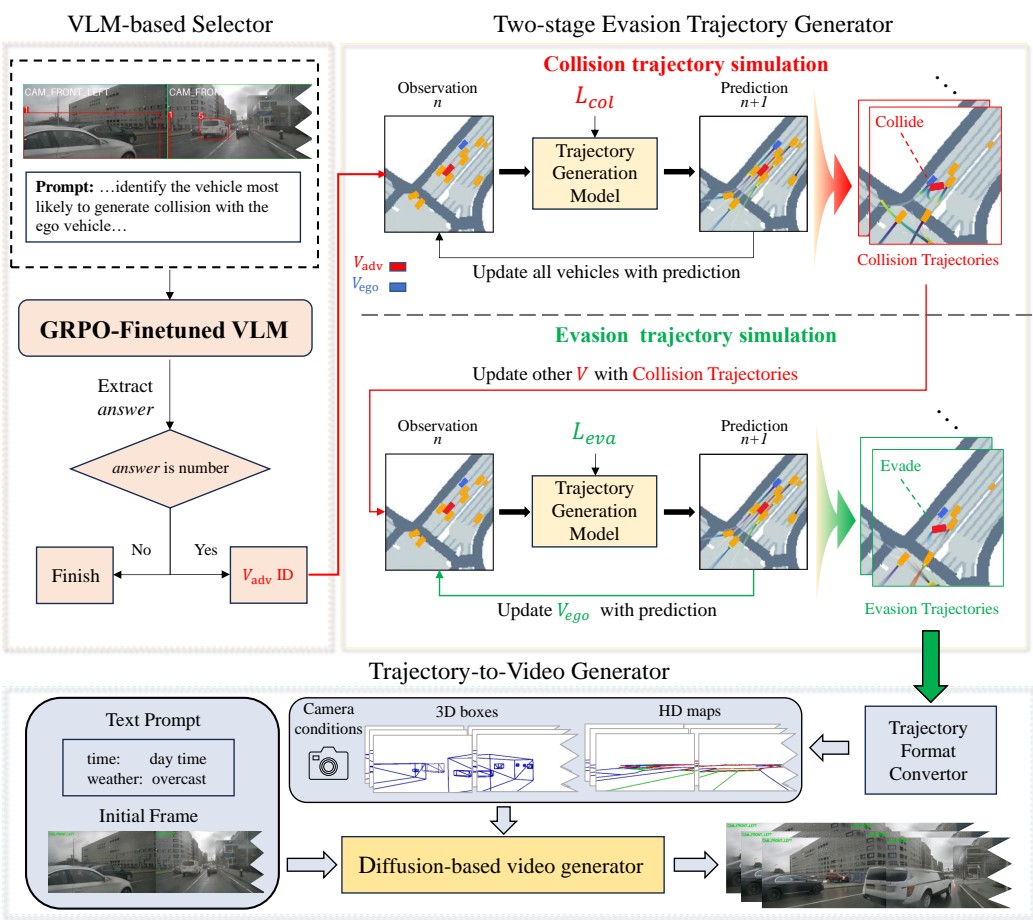

Figure 2: The SafeMVDrive framework for generating realism, multi-view safety-critical videos.

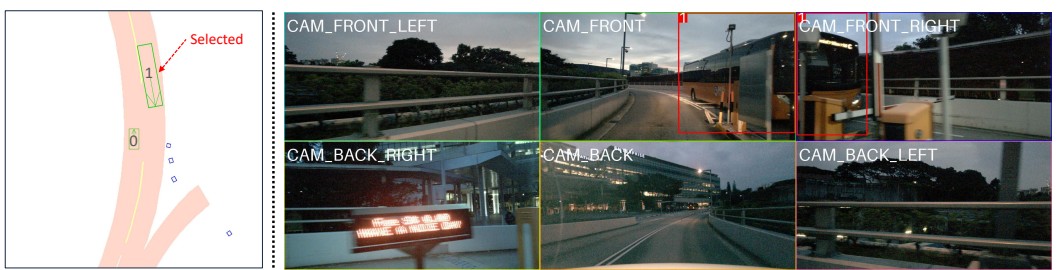

Figure 3: Comparison between the initial scene's BEV-rendered annotations (left) and its camera images (right). Previous selection methods rely on non-visual annotations, which would select vehicle 1. However, in the real scene, an obstacle exists between the ego vehicle and vehicle 1, preventing a physically feasible collision.

and accurate vehicle ID output, we annotate safety-critical vehicle candidates with ID-labeled 2D bounding boxes (as shown in the right part of Figure 3) and exclude distant vehicles (beyond distance $D$ from the ego vehicle). Our first attempt is to guide the VLM with task-specific prompts (see Appendix A.4). However, due to VLMs' limited exposure to multi-view data during training, prompting alone proves insufficient for effective multi-view understanding.

To address the above problem, we propose a VLM fine-tuning pipeline to improve its adaptation to our task. A key challenge in constructing a suitable fine-tuning dataset is determining the correct VLM

output for each multi-view image—specifically, identifying which vehicles could realistically collide with the ego vehicle via dynamically plausible trajectories. To this end, we propose an automated method that simulates each safety-critical vehicle candidate using a dynamically plausible traffic simulator (Zhong et al., 2023b) to examine whether it can collide with the ego vehicle. Specifically, for each candidate, we apply a loss-guidance encouraging collision with the ego vehicle based on their distance (see Section 3.3). We select vehicles that successfully collide and filter out unrealistic cases, such as those entering non-drivable areas or colliding with other vehicles first. This yields annotated data defining the set of effective safety-critical vehicles $S_{coll}$ for each scene. Because repeated simulation is computationally expensive, the method is used only during training to construct reliable fine-tuning labels, while at inference, the VLM performs fast selection without exhaustive simulation.

After obtaining the fine-tuning dataset, we apply the GRPO algorithm (Shao et al., 2024), a recent RL method proven to effectively enhance the reasoning of VLMs (Shen et al., 2025). GRPO enhances the model's reasoning capabilities through a self-improving RL process, which makes it well-suited for helping the model better understand complex multi-view physical scenarios (Wei et al., 2022; Ho et al., 2022). Following (Shen et al., 2025), we augment the prompt with: 'Output the thinking process in `<think> </think>` and final answer (number) in `<answer> </answer>` tags.' and use the following format reward:

$$R_{form} = \begin{cases} 1 & \text{if } O \sim \texttt{<think>...</think> <answer>...</answer>} \\ 0 & \text{otherwise} \end{cases} \quad (1)$$

where $O$ refers to the output from the VLM. The format reward is designed to enforce the model to place its reasoning process between '`<think>`', and outputs the final answer within the '`<answer>`' tags. To promote accurate outputs, we add the following accuracy reward,

$$R_{Acc} = \begin{cases} similarity(extract\_answer(O), \text{"no vehicle is appropriate"}) & \text{if } S_{\text{coll}} = \emptyset \\ 1 & \text{if } S_{\text{coll}} \neq \emptyset \wedge extract\_answer(O) \in S_{\text{coll}} \\ 0 & \text{otherwise} \end{cases} \quad (2)$$

where $extracted\_answer()$ extracts the content between the tags `<answer>...</answer>` from the VLM output. $S_{\text{coll}}$ denotes the set of vehicles that can collide with the ego vehicle in our automated annotation process. If $S_{\text{coll}} = \emptyset$, it indicates that no vehicle in the scene can collide with the ego vehicle, in which case we expect the model to output "no vehicle is appropriate". If $S_{\text{coll}} \neq \emptyset$, at least one adversarial vehicle exists, and the model should output an ID belonging to this set.

### 3.3 TWO-STAGE EVASION TRAJECTORY GENERATOR

After selecting the adversarial vehicles as described above, current safety-critical trajectory simulators (Xu et al., 2025; Zhong et al., 2023a; Chang et al., 2024) can be used to generate collision trajectories. However, current multi-view video generators struggle to realistically generate such collision events, resulting in degraded visual quality when control signals cause collisions. To address this, we propose a two-stage evasion trajectory generator. It produces safety-critical yet non-colliding evasion trajectories compatible with current video generators while retaining safety-critical features.

Our generator builds upon a popular diffusion-based controllable trajectory generation framework (Zhong et al., 2023b). The model is first trained on real-world driving data to learn realistic trajectories. Since our framework takes a single-frame initial scene as input, we retrain the trajectory generation model to align with this setup. At test time, a loss-gradient guidance (Janner et al., 2022; Liang et al., 2023; Chang et al., 2024; Zhong et al., 2023b) is introduced to enable controllable trajectory generation. This guidance pushes the sampled trajectories toward the data manifold of realistic trajectories while simultaneously minimizing the loss function. We adopt the closed-loop simulation strategy described in (Zhong et al., 2023b). At each step $n$, the model predicts future trajectories from the current scene, applying only the first few to update the scene. This iterates to form the full trajectory sequence.

Once the VLM-based selector identifies the adversarial vehicle $V_{adv}$, our trajectory generator performs a two-stage simulation. In the first stage, $V_{adv}$ is guided to collide with $V_{ego}$. If the collision occurs before $V_{adv}$ entering non-drivable areas or hitting others, the collision trajectory is considered valid.

Table 1: Comparison of baseline effectiveness by evaluating video realism and planner's safety-related metrics on the videos. Sample-level $CR$ measures the average collision rate per valid sample, while scene-level $CR$ counts the average number of colliding valid samples per scene. SafeMVDrive has a natural score comparable to Origin (0.9x), with a 15x CR increase and 30% reductions in TTC and NC.

| Methods | Natural score ↑ | Subject Consistency↑ | FID ↓ | Sample-level $CR$ (%) ↑ | | | | Scene-level $CR$ ↑ | | | | NC↓ | TTC↓ |
|---|---|---|---|---|---|---|---|---|---|---|---|---|---|
| | | | | 1s | 2s | 3s | Avg. | 1s | 2s | 3s | Avg. | | |
| Origin | $0.63 \pm 0.13$ | 0.88 | 16.25 | 0.57 | 0.90 | 1.64 | 1.04 | 0.07 | 0.12 | 0.21 | 0.13 | 0.98 | 0.96 |
| Naive | $0.21 \pm 0.18$ | 0.66 | 23.35 | 18.65 | 34.04 | 46.40 | 33.03 | 1.46 | 2.66 | 3.62 | 2.58 | 0.37 | 0.28 |
| SafeMVDrive | $0.56 \pm 0.12$ | 0.86 | 20.63 | 6.77 | 15.25 | 23.67 | 15.23 | 0.86 | 1.94 | 3.02 | 1.94 | 0.77 | 0.69 |

In the second stage, we introduce a trajectory update mechanism with an evasion-targeted loss, which guides $V_{ego}$ in evading $V_{adv}$. Finally, a collision-evasion trajectory sequence is generated.

During the collision-stage trajectory simulation, we employ three loss functions for test-time guidance: an adversarial loss, a no-collision loss, and an on-road loss. The adversarial loss is necessary to encourage $V_{adv}$ to collide with $V_{ego}$, typically based on their distance (Zhong et al., 2023a; Chang et al., 2024). However, this often causes $V_{adv}$ to remain stuck to $V_{ego}$ after collision, resulting in unnatural dynamics—shifting of $V_{adv}$ from aggressive (e.g., rapid acceleration) to passive, ego-like behaviors (e.g., slow driving). To solve this, we propose the following adversarial loss formulation:

$$L_{adv} = \begin{cases} \sum_{t=1}^{T} w_t \cdot d_t \cdot \mathbb{I}(d_t > d_{penalty}) & \text{Before } V_{adv} \text{ collides with } V_{ego} \\ 0 & \text{After } V_{adv} \text{ collides with } V_{ego} \end{cases} \tag{3}$$

where $T$ denotes predicted future steps, $d_t$ is the distance between $V_{adv}$ and $V_{ego}$ at time step $t$, and $d_{\text{penalty}}$ is the non-collision distance threshold. Gradients are detached with respect to $V_{ego}$ to ensure only $V_{adv}$ has the adversarial behavior. A time-decay weight $w_t = \frac{\lambda^t}{\sum_{k=0}^{T-1} \lambda^k}$, controlled by a decay factor $\lambda$, emphasizes earlier trajectory predictions and is shared across all losses. Moreover, to avoid unnatural sticking post-collision, we explicitly set the adversarial loss $L_{adv}$ to zero once the collision between $V_{adv}$ and $V_{ego}$ has occurred in the updated trajectories during closed-loop simulation. This leads to more natural post-collision behavior of the adversarial vehicle.

To prevent undesired collisions (except between the ego and adversarial vehicles), we utilize a no-collision loss $L_{no\_coll}$, which penalizes inter-vehicle collisions in denoised trajectories, excluding the ego–adversarial pair. To keep vehicles in drivable areas, an on-road loss $L_{on\_road}$ penalizes trajectories entering non-drivable zones and guides them back. Full definitions are in Appendix A.3.

Overall, the loss function of the collision stage trajectory simulation can be summarized as

$$L_{coll} = \alpha L_{adv} + \beta L_{no\_coll} + \gamma L_{on\_road} \tag{4}$$

where $\alpha$, $\beta$, $\gamma$ are the hyperparameters to control the contribution of each loss function. Through closed-loop simulation, we obtain a trajectory sequence. To ensure safety-criticality and physical feasibility, we filter out trajectories that either do not result in a collision with the ego vehicle or that collide with other vehicles or go off-road before reaching the ego vehicle. Subsequently, the physically feasible and safety-critical trajectories are fed into the second stage for evasion trajectory simulation.

During the evasion stage trajectory simulation, we only use $L_{no\_coll}$ and $L_{on\_road}$ as follow,

$$L_{eva} = \beta L_{no\_coll} + \gamma L_{on\_road} \tag{5}$$

where the $L_{no\_coll}$ is applied to all vehicles in the scene (include ego–adversarial pair) to guide $V_{ego}$ to evade $V_{adv}$. The evasion-stage simulation starts from the same initial scene as the collision stage. During closed-loop rollout, only the ego vehicle's trajectory is updated via the diffusion model; other vehicles retain their collision-stage trajectories to preserve adversarial behavior. This converts a collision scenario to a safety-critical evasion one, staying within the video generator's capability. Finally, we select successful evasive trajectories for video generation.

Figure 4: Comparison of videos generated by different methods, only showing the front view. It can be seen that the video generated by the origin method is quite ordinary, the video generated by the naive method loses realism towards the end (highlighted by the red-boxed frames), while only ours exhibits both realism and safety-criticality.

## 3.4 TRAJECTORY-TO-VIDEO GENERATOR

To convert simulated collision-evasion trajectories into multi-view driving videos, we adopt UniM-LVG (Chen et al., 2024b), a diffusion-based video generation model tailored for autonomous driving. UniM-LVG is trained on 1,498 hours of diverse driving data (including OpenDV-Youtube, NuScenes, Waymo, and Argoverse2), providing strong out-of-distribution generalization that enables it to accurately represent our safety-critical collision-avoidance scenarios. This model supports motion control signals such as 3D bounding boxes, HD maps, and camera conditions, enabling trajectories to be translated into realistic multi-view videos. Moreover, UniMLVG can produce high-quality and sufficiently long videos, which is essential since safety-critical events typically unfold over longer time spans, by mitigating autoregressive errors through its multi-task training scheme. Detailed generation settings are provided in Appendix A.9.

## 4 EXPERIMENTS

### 4.1 EXPERIMENTAL SETTINGS

**Datasets:** We use the large-scale real-world driving dataset NuScenes (Caesar et al., 2020), which provides 5.5 hours of annotated trajectories collected across two cities, featuring diverse scenarios and traffic conditions. To train the VLM for our adversarial vehicle selector, we randomly select 1,500 samples from the training split and generate the safety-critical annotations within each scene with the automated annotation method (see Section 3.2). The trajectory generation diffusion model is trained on the full training split. For evaluation, 250 samples are randomly selected from the validation split.

**Baseline:** As the first to generate multi-view, realistic, safety-critical videos, we design two intuitive baselines for comparison. **Naive** generates safety-critical videos by converting collision trajectories into control signals for the video generator. These trajectories are produced with the vehicle selection method and loss function from (Zhong et al., 2023a), combined with our retrained trajectory diffusion model. **Origin** uses original NuScenes trajectories to benchmark video quality under natural trajectories. All baselines generate videos via UniMLVG with identical settings (see Appendix A.9).

**Metrics:** To evaluate the realism of generated videos, existing automatic metrics such as FVD (Unterthiner et al., 2018) are widely recognized as insufficient for accurately reflecting perceptual quality and real-world dynamics (Gao et al., 2024b; Bar-Tal et al., 2024; Girdhar et al., 2024; Wu et al., 2024). Consequently, we rely on human evaluation for a more reliable assessment. In line with recent studies (Gao et al., 2024b; Bar-Tal et al., 2024; Blattmann et al., 2023b;a; Chen et al., 2023; Wang et al., 2025), we employ the Two-Alternative Forced Choice (2AFC) protocol to evaluate the videos (see Appendix A.2 for details) and refer to the resulting preference rate as the **natural score** in our experiments. In addition, we also compute FID, an auxiliary metric for evaluating image quality.

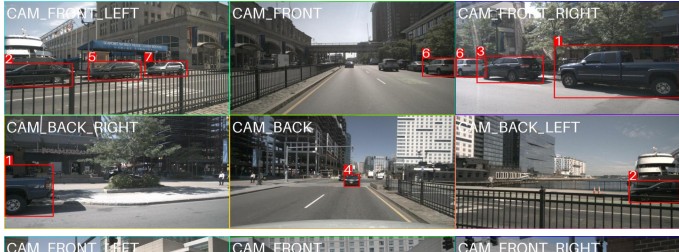

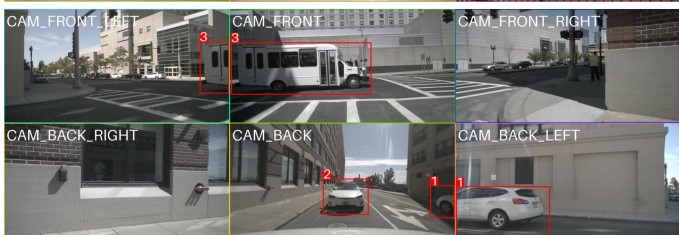

Figure 5: Adversarial vehicle selection examples with GRPO-finetuned VLM. Images are inputs; texts are responses. The VLM correctly analyzes the vehicle's position and the visual information fence.

To evaluate physical consistency, we introduce Subject Consistency. Intuitively, stable tracking reflects physical plausibility, as tracking often fails when an object deforms unrealistically. Since our scenarios focus on collision avoidance between the ego vehicle and an adversarial vehicle, the adversarial vehicle is most likely to exhibit such inconsistencies. We therefore use StreamPETR (Wang et al., 2023) to track the adversarial vehicle in the generated videos and define Subject Consistency as the ratio between its tracked frame length and its actual frame length in the scene. For Origin, we use the same adversarial (AD) vehicle ID as in SafeMVDrive.

To demonstrate videos generated by our framework present significant challenges to end-to-end planners and are likely to induce collisions, we evaluate three end-to-end planners, UniAD (Hu et al., 2023), SparseDrive (Sun et al., 2025), and DiffusionDrive (Liao et al., 2025) on our generated data. Following the method proposed by Li et al. (Li et al., 2024b), we compute the collision rate for each video sample. Specifically, the collision rate is defined as:

$$cr(t) = \left(\sum_{i=0}^{N} \mathbb{I}_i\right) > 0, \quad N = \frac{t}{0.5}.$$

(6)

where $N$ denotes the number of the trajectory points in the planning before $t$ seconds, and $\mathbb{I}_i$ indicates whether the ego vehicle collides at step $i$. The final collision rate $CR(t)$ is averaged over all samples. We follow this method with one adaptation: if a collision already occurs at the initial step ($\mathbb{I}_0 = 1$), the collision ratio cr(t) for this sample is always 1, making it unsuitable for evaluating planner performance. Therefore, we consider such samples as invalid initializations and exclude them from our evaluation. We report both sample-level and scene-level averaged $CR(t)$: the former represents the averaged $cr(t)$ before time t for each valid sample; the latter reflects the average number of valid samples in which the planner collides before time $t$ within each scene. For a more comprehensive evaluation, we follow NAVSIM (Dauner et al., 2024), which benchmarks planning using non-reactive simulations and closed-loop metrics. We focus on two collision-related measures, no collisions (NC) and time-to-collision (TTC), re-implemented under our datasets (see Appendix A.11).

**Implement Details**: We use Qwen2.5VL-7B-Instruct (Team, 2025) as the base VLM for its strong vision-language understanding capabilities (fine-tuning details in Appendix A.8). For safety-critical candidate selection, we set the distance threshold $D = 25$ m. Moreover, we retrain the trajectory generation model to align with our single-frame input setup (details in Appendix A.8). In the two-stage simulation process, we set $\alpha = 1$, $\beta = 50$, $\gamma = 1$ in the collision stage, $\beta = 1$, $\gamma = 1$ in the evasion stage, and use $\lambda = 0.9$ for all loss terms. Ablation studies on these hyperparameters can be found in Appendix A.14 and A.15. Our video generator produces 19 frames per iteration, using the last frame of the previous rollout as the reference frame for the next. This results in a final 9-second video at 12Hz. Efficiency and cost analysis of our framework can be found in Appendix A.10.

Table 2: Comparison of different methods for adversarial vehicle selection.

| Methods | Precision | Recall | F1-score |
|---------|-----------|--------|----------|
| Random-neighbor rule | 0.606 | 0.437 | 0.507 |
| Precision-first rule | 0.758 | 0.497 | 0.600 |
| Nearest-vehicle rule | 0.528 | 0.861 | 0.654 |
| VLM-based selector | 0.750 | 0.675 | **0.710** |

Table 3: Comparison of the performance of different models on adversarial vehicle selection.

| Model | Precision | Recall | F1-score |
|-------|-----------|--------|----------|
| Base 3B | 0.360 | 0.596 | 0.449 |
| Base 7B | 0.455 | 0.530 | 0.489 |
| Base 72B | 0.433 | 0.602 | 0.504 |
| SFT-finetuned Model 7B | 0.582 | 0.748 | 0.655 |
| GRPO-fine-tuned Model 7B | 0.750 | 0.675 | **0.710** |

Table 4: Evaluation of the effectiveness of the two-stage simulation.

| Methods | Natural score ↑ | Subject Consistency↑ | FID ↓ | Sample-level $CR$ (%) ↑ | | | | Scene-level $CR$ ↑ | | | | NC↓ | TTC↓ |
|---------|-----------------|----------------------|-------|------|------|------|------|------|------|------|------|------|------|
| | | | | 1s | 2s | 3s | Avg. | 1s | 2s | 3s | Avg. | | |
| Origin | 0.60 ± 0.13 | 0.88 | 16.25 | 0.57 | 0.90 | 1.64 | 1.04 | 0.07 | 0.12 | 0.21 | 0.13 | 0.98 | 0.96 |
| Collision stage only | 0.33 ± 0.13 | 0.58 | 22.20 | 17.53 | 31.15 | 38.98 | 29.22 | 1.46 | 2.59 | 3.24 | 2.43 | 0.67 | 0.63 |
| Two-stage simulation | 0.57 ± 0.08 | 0.86 | 20.63 | 6.77 | 15.25 | 23.67 | 15.23 | 0.86 | 1.94 | 3.02 | 1.94 | 0.77 | 0.69 |

## 4.2 EVALUATION OF SAFEMVDRIVE

This section compares the realism of videos generated by different baselines, as well as the averaged safety-critical-related metric scores (CR, NC, and TTC) of UniAD, SparseDrive, and DiffusionDrive on these videos (respective detailed results presented in the Appendix). Each method generates videos from 250 samples randomly selected from the NuScenes validation split (details in Appendix A.13); the results are shown in Table 1. For video generation, naturalness is the foundation of usability. SafeMVDrive achieves both higher naturalness and FID scores than the Naive baseline. In particular, its naturalness score is nearly three times higher than that of Naive and remains comparable to Origin. The extremely low naturalness score of Naive is mainly attributed to excessive collision events exceeding the generator's capacity and leading to severe vehicle deformations (as shown in red-boxed frames in Fig. 4; video examples are provided here ). Moreover, while preserving naturalness and image quality, SafeMVDrive introduces more realistic safety-critical events, which in turn challenge the planner, resulting in a substantial increase in CR and significant decreases in TTC and NC compared to Origin.

## 4.3 EVALUATION OF THE VLM-BASED ADVERSARIAL VEHICLE SELECTOR

In this section, we evaluate the effectiveness of our VLM-based adversarial vehicle selector. On 250 validation scenes, we identify all vehicles in the simulated scenes that may collide with the ego vehicle by exhaustively checking every vehicle in the environment (unlike in VLM fine-tuning label generation, where distant vehicles are filtered out). We compare the precision, recall, and F1-score of our VLM-based selector against three heuristic methods: Random-neighbor rule (Chang et al., 2024), Precision-first rule (Zhong et al., 2023a), and Nearest-vehicle rule (details in Appendix A.12). As shown in Table 2, our method achieves the highest F1-score, demonstrating its effectiveness in accurately identifying safety-critical vehicles. Figure 5 shows examples of adversarial vehicle selections, where our VLM correctly analyzes positional relationships and driving directions to make appropriate selections.

We further evaluate the effectiveness of our GRPO fine-tuning. As shown in Table 3, the GRPO-fine-tuned model significantly outperforms the untuned baseline, achieving an F1-score improvement of 0.21 over the strongest 72B base model. We also evaluate supervised fine-tuning (SFT) for comparison (see Appendix A.8 for configurations), but it performs worse than GRPO, with an F1-score reduction of more than 0.05. These findings highlight both the necessity and effectiveness of adopting GRPO for our adversarial vehicle selection task.

## 4.4 EVALUATION OF THE EFFECTIVENESS OF THE TWO-STAGE SIMULATION

We propose a two-stage trajectory simulator to generate collision-evasion scenarios that are both safety-critical and within the capability of current multi-view video generators. In this section, we assess the necessity of the two-stage simulation by comparing videos from collision-stage-only

Table 5: Effectiveness of SafeMVDrive in improving the performance of the End-to-end planner.

| Evaluation Set | Model | Sample-level $CR$ (%) $\downarrow$ | | | | Scene-level $CR \downarrow$ | | | | NC$\uparrow$ | TTC$\uparrow$ |
|---|---|---|---|---|---|---|---|---|---|---|---|
| | | 1s | 2s | 3s | Avg. | 1s | 2s | 3s | Avg. | | |
| NuScenes Val | Base | 0.11 | 0.15 | 0.28 | 0.18 | 0.04 | 0.05 | 0.09 | 0.06 | 0.992 | 0.976 |
| | Finetuned | 0.06 | 0.11 | 0.21 | 0.13 | 0.02 | 0.04 | 0.07 | 0.04 | 0.993 | 0.979 |
| SafeMVDrive Val | Base | 6.63 | 17.60 | 25.68 | 16.64 | 0.81 | 2.16 | 3.15 | 2.04 | 0.743 | 0.684 |
| | Finetuned | 3.06 | 7.40 | 12.59 | 7.68 | 0.37 | 0.91 | 1.54 | 0.94 | 0.822 | 0.763 |

trajectories with those from the full two-stage process. Additionally, to assess the naturalness of the generated videos, we include the Origin baseline for comparison. Each method generates videos based on 250 samples randomly selected from the NuScenes validation split. Detailed generation procedures can be found in Appendix A.13. As shown in Table 4, our two-stage simulation leads to both higher collision rates for planners and significantly improved realism.

### 4.5 ENHANCEMENT FOR END-TO-END PLANNER

In this section, we verify the usefulness of the videos generated by SafeMVDrive for improving an end-to-end autonomous driving planner in terms of collision avoidance. We split the generated data into training and validation sets with a 4:1 ratio, and mix the training portion with the original NuScenes training set to fine-tune the lightweight DiffusionDrive (Liao et al., 2025) model for 10 epochs. We then evaluate the model on both the NuScenes validation set and our generated SafeMVDrive validation set. As shown in Table 5, after training the end-to-end planner on a combination of our generated data and the real NuScenes data, the collision-related metrics decrease significantly on the SafeMVDrive validation dataset, and also decrease on NuScenes validation dataset.

Building on these results, we note that the two-stage avoidance scenario simulation allows our generated data to contain active collision-avoidance behaviors, which provides a richer supervisory signal for planning. As a result, the end-to-end planner can effectively learn these behaviors and reduce collision rates. This further demonstrates the value of our framework.

## 5 CONCLUSION

We present **SafeMVDrive**, the first framework for generating multi-view safety-critical driving videos in the real-world domain. By strategically combining a safety-critical trajectory simulator with a realistic multi-view video generator, we build a bridge from safety-critical trajectory simulation to multi-view video generation. To address the integration challenge, we introduce a VLM-based adversarial vehicle selector and a two-stage collision-evasion trajectory generation strategy. Experiments demonstrate the effectiveness of our approach in producing realistic and multi-view safety-critical videos, which lead to a high collision rate for end-to-end planners. The generated video data can serve as valuable resources for evaluating and enhancing autonomous driving systems.

## 6 REPRODUCIBILITY STATEMENT

We have made extensive efforts to ensure the reproducibility of our work. The source code of our framework is available at https://anonymous.4open.science/r/SafeMVDrive_anonymous_code-D70E. The overall design and methodology of our framework are described in Section 3 of the main text, while detailed training and evaluation settings are provided in the Appendix. In addition, we will release our test dataset to the public upon acceptance of the paper.

## 7 ETHICS STATEMENT

This work aims to improve the safety and robustness of autonomous driving systems by generating realistic, safety-critical driving scenarios for testing and training. The potential for misuse is limited, as the primary application—autonomous driving—rarely involves malicious intent.

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

# A  APPENDIX

## A.1  DISCUSSION ABOUT DIFFERENT TYPES OF SAFETY-CRITICAL SCENARIOS

In autonomous driving scenarios, the most severe type of incident is vehicle-to-vehicle collisions, which can lead to serious threats to human safety. In the past, constructing datasets of inter-vehicle collision scenarios has been an important research topic (Wang et al., 2024; Chang et al., 2024; Wang et al., 2021). However, it remains difficult to obtain multi-view collision scenes in the real world domain. Therefore, the main goal of this work is to generate multi-view, safety-critical scenarios involving vehicle-to-vehicle collision avoidance in real-world domains.

Generating a broader variety of safety-critical scenarios would be even more beneficial for autonomous driving. Our current framework focuses on safety-critical collision-avoidance scenarios between the ego vehicle and an adversarial vehicle. Since safety-critical situations are highly diverse, we next discuss how our approach could be extended to two additional common types of scenarios. (1): for collisions avoidance between two non-ego vehicles, one only need to replace the loss-guidance target for the adversarial vehicle with the other vehicle in the collision stage simulation. (2): for safety-critical interactions between pedestrians and vehicles, one need to train a diffusion-based trajectory generator specifically for pedestrians and control their motion using a test-time loss-guidance strategy similar to that proposed in our paper. Furthermore, existing autonomous driving video generators still exhibit inadequate performance when conditioned on out-of-distribution pedestrian trajectories, indicating that stronger modeling of pedestrians is required to ensure high-quality generation in such scenarios.

## A.2  USER STUDY SETTING

In our experiments, participants are presented with two videos displayed side-by-side and are asked to choose the one they perceive to be of higher visual quality. In addition to choosing one of the two videos, an 'uncertain' option is also provided. A selected video receives 1 point; in the case of an "uncertain" response, both videos receive 0.5 points each. The final realism score is computed as the total number of points received divided by the total number of comparisons. The questionnaire we used is shown in Figure 6. The user studies in Section 4.2 and Section 4.4 are conducted separately. In the experiment of Section 4.2, we randomly select ten initial scenes that are present across all three video sets—Origin, Naive, and SafeMVDrive. For each selected scene, we retrieve the corresponding video from each set, forming ten matched triplets for pairwise comparison. Similarly, in Section 4.4, we randomly select ten initial scenes that exist in all three sets—Origin, Collision Stage Only, and Two-Stage Simulation—and obtain the corresponding video per method for each scene, again resulting in ten matched triplets for evaluation. For each user study, we collect 660 answers from 22 participants.

## A.3  NO-COLLISION LOSS AND ON-ROAD LOSS

To prevent collisions between the vehicles in the scene, we use the following no-collision loss,

$$L_{no\_coll} = \sum_{t=1}^{T} \sum_{i,j \in \mathcal{A}} w_t \cdot \left(1 - \frac{d_t^{i,j}}{d_{penalty}^{i,j}}\right) \cdot M_{i,j} \cdot \mathbb{I}((d_t^{i,j} < d_{penalty}^{i,j}) \wedge (v_i > v_{th})) \quad (7)$$

where $\mathcal{A}$ is the set of all vehicles in the scene, and $d_t^{i,j}$ and $d_{penalty}^{i,j}$ represent the distance at time step $t$ and minimum non-collision threshold distance while detach the gradient of $V_j$. $v_i$ is the velocity of the vehicle $V_i$, and $v_{th}$ is a very small velocity threshold. When the vehicle's velocity exceeds $v_{th}$, it indicates that the vehicle is not in a completely stationary state. This condition ensures that when a moving vehicle is about to collide with a stationary vehicle, the moving vehicle will adjust its trajectory, rather than causing the stationary vehicle to evade the collision, which is a more natural way to prevent collisions. $M_{i,j}$ is a mask indicating which pairs of agents should evade collisions. In the collision-stage simulation, we configure that all agents, except for the ego and adversarial vehicles, are required to evade collisions. In the evasion-stage simulation, this mask includes all agents.

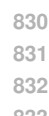
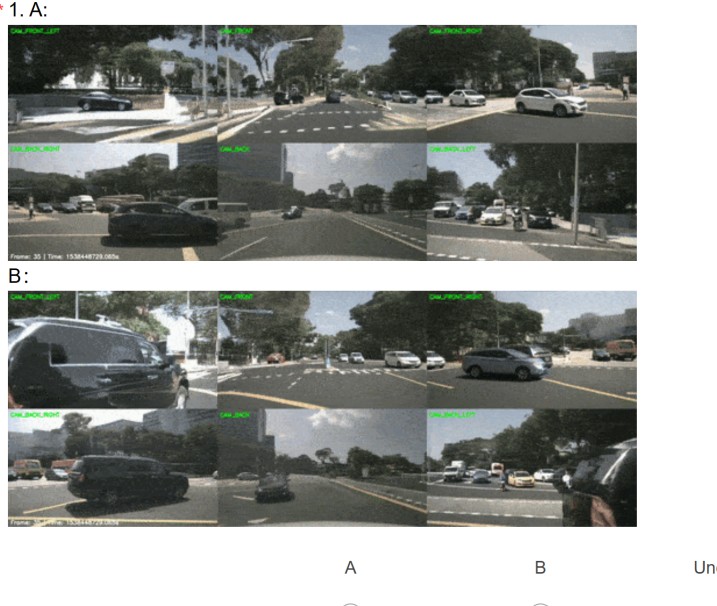

**Realism Evaluation of Autonomous Driving Videos**

Evaluate the realism of multi-view videos from the ego-vehicle perspective. Please select the option that appears more realistic in each video comparison.

\* 1. A:

B:

|  | A | B | Uncertain |
|---|---|---|---|
| The more realistic video is | ○ | ○ | ○ |

Figure 6: The questionnaire used to evaluate the realism of videos generated by different baselines in the user study.

To ensure that the vehicles stay within the driving area, we utlilize the following on-road loss,

$$L_{on\_road} = \sum_{t=1}^{T} \sum_{p \in P_{offroad}} w_t \cdot \left(1 - \frac{\min_{q \in P_{onroad}} dist(p,q)}{l_{diag}}\right) \cdot \mathbb{I}(v_i > v_{th}) \tag{8}$$

where $P_{offroad}$ and $P_{onroad}$ are the set of sampled points located off-road and on-road, respectively, within the agent vehicle's bounding box, $dist(p,q)$ is the Euclidean distance between points $p$ and $q$ while detach the gradients of $q$, and $l_{diag}$ is the diagonal length of the agent vehicle's bounding box. The gradient of this loss pulls the ego vehicle's off-road points toward the nearest on-road points, thereby encouraging the denoised trajectory to remain within drivable areas.

### A.4 PROMPT

In this section, we present the prompt used in our VLM-based selection of adversarial vehicles in Figure 10. In the prompt, $v$ is substituted with the ego vehicle's velocity in the given scene. For the non-finetuned VLM, we append "Output final answer (number) in `<answer> </answer>` tags." at the end to ensure it outputs the vehicle ID for evaluation. For the GRPO-fine-tuned VLM, we append "Output the thinking process in `<think> </think>` and final answer (number) in `<answer> </answer>` tags." at the end. For the SFT-finetuned VLM, we use the original prompt without any modifications.

Table 6: Performance of UniAD in Section 4.2

| Methods | Sample-level $CR$ (%) ↑ | | | | Scene-level $CR$ ↑ | | | | NC↓ | TTC↓ |
|---|---|---|---|---|---|---|---|---|---|---|
| | 1s | 2s | 3s | Avg. | 1s | 2s | 3s | Avg. | | |
| Origin | 1.14 | 2.09 | 3.98 | 2.40 | 0.146 | 0.27 | 0.51 | 0.31 | 0.99 | 0.97 |
| Naive | 19.04 | 43.59 | 61.57 | 41.40 | 1.49 | 3.40 | 4.81 | 3.23 | 0.33 | 0.26 |
| SafeMVdrive | 2.10 | 11.85 | 23.52 | 12.49 | 0.27 | 1.51 | 3.00 | 1.59 | 0.83 | 0.75 |

Table 7: Performance of SparseDrive in Section 4.2

| Methods | Sample-level $CR$ (%) ↑ | | | | Scene-level $CR$ ↑ | | | | NC↓ | TTC↓ |
|---|---|---|---|---|---|---|---|---|---|---|
| | 1s | 2s | 3s | Avg. | 1s | 2s | 3s | Avg. | | |
| Origin | 0.38 | 0.38 | 0.50 | 0.42 | 0.05 | 0.05 | 0.07 | 0.05 | 0.98 | 0.96 |
| Naive | 17.26 | 28.07 | 38.14 | 27.82 | 1.35 | 2.19 | 2.98 | 2.17 | 0.38 | 0.29 |
| SafeMVdrive | 7.57 | 15.23 | 22.32 | 15.04 | 0.96 | 1.94 | 2.84 | 1.91 | 0.74 | 0.67 |

## A.5 EVALUATION RESULTS OF DIFFERENT END-TO-END PLANNERS

In this section, we present the detailed open-loop performance of the different planners evaluated in Sections 4.2 and 4.4 of the main paper. Tables 6, 7, and 8 report the detailed results of UniAD (Hu et al., 2023), SparseDrive (Sun et al., 2025), and DiffusionDrive (Liao et al., 2025) for the experiments in Section 4.2. Tables 9, 10, and 11 show the corresponding results for the experiments in Section 4.4.

## A.6 EXAMPLES OF VLM SELECTING PHYSICALLY FEASIBLE AND SAFETY-CRITICAL VEHICLES

In this section, we present several examples in which the VLM makes selections that adhere to physical feasibility, as shown in Figures 7, 8, and 9. In these examples, the vehicles selected by the most representative rule based method, nearest-vehicle rule method, are in fact physically infeasible, whereas our VLM makes physically feasible choices.

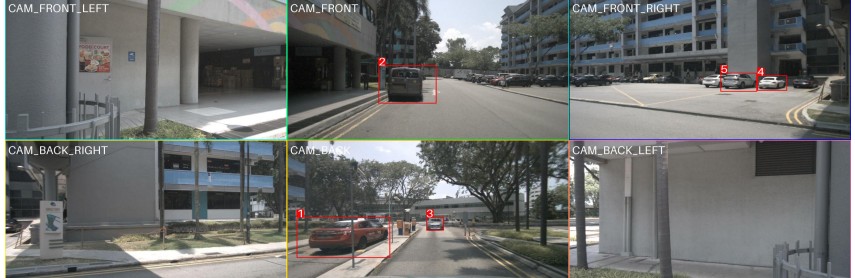

Figure 7: In this example, the nearest-vehicle rule method would choose Vehicle 1, which is blocked by an obstacle and therefore cannot produce a physically feasible scenario, whereas the VLM would select Vehicle 3.

## A.7 RUNNING TIME COMPARISON BETWEEN EXHAUSTIVE SIMULATION AND VLM INFERENCE.

In this section, we compare the computational efficiency of our VLM-based selector against exhaustive simulation. Since trajectory simulation incurs a non-trivial computational cost—approximately 2 minutes per simulation—simulating every vehicle in a scene to identify safety-critical ones is time-

Table 8: Performance of DiffusionDrive in Section 4.2

| Methods | Sample-level $CR$ (%) ↑ | | | | Scene-level $CR$ ↑ | | | | NC↓ | TTC↓ |
|---|---|---|---|---|---|---|---|---|---|---|
| | 1s | 2s | 3s | Avg. | 1s | 2s | 3s | Avg. | | |
| Origin | 0.19 | 0.24 | 0.44 | 0.29 | 0.02 | 0.03 | 0.06 | 0.04 | 0.98 | 0.97 |
| Naive | 19.66 | 30.47 | 39.50 | 29.88 | 1.54 | 2.39 | 3.09 | 2.34 | 0.41 | 0.30 |
| SafeMVdrive | 10.63 | 18.68 | 25.17 | 18.16 | 1.35 | 2.38 | 3.21 | 2.31 | 0.74 | 0.66 |

Table 9: Performance of UniAD in Section 4.4

| Methods | Sample-level $CR$ (%) ↑ | | | | Scene-level $CR$ ↑ | | | | NC↓ | TTC↓ |
|---|---|---|---|---|---|---|---|---|---|---|
| | 1s | 2s | 3s | Avg. | 1s | 2s | 3s | Avg. | | |
| Origin | 1.14 | 2.09 | 3.98 | 3.98 | 0.15 | 0.27 | 0.51 | 0.31 | 0.99 | 0.97 |
| Collision stage only | 20.35 | 40.12 | 48.26 | 48.26 | 1.71 | 3.37 | 4.05 | 3.04 | 0.67 | 0.64 |
| Two-stage simulation | 2.10 | 11.85 | 23.52 | 23.52 | 0.27 | 1.51 | 3.00 | 1.59 | 0.83 | 0.75 |

consuming, typically averaging around 28 minutes and exceeding 98 minutes in scenes with many vehicles. Even when used purely as part of a data engine, such simulation time introduces significant overhead. In contrast, once trained, the VLM requires only about 22 seconds per scene to accurately identify safety-critical adversarial vehicles, yielding roughly an 80× speed-up. Moreover, according to the scale laws of autonomous driving (Naumann et al., 2025), training a commercial planner generally requires massive datasets, often amounting to tens of thousands of hours of driving data. At such a scale, the speed of data generation becomes crucial, as it can directly influence the overall time-to-market.

## A.8 FINETUNING SETTING

### A.8.1 VLM FINETUNING

**GRPO-finetuning details**: We set the learning rate to 0.00002 with a cosine scheduler, enable DeepSpeed Zero3, set the number of generations in GRPO to 6, do not freeze any modules, and follow other settings from the LoRA fine-tuning configuration in (Shen et al., 2025). We fine-tune Qwen-VL 2.5 Instruct (Team, 2025) using the GRPO algorithm within the framework of (Shen et al., 2025) for 2600 steps on 4 A800 GPUs.

**SFT-finetuing details**: We set the learning rate to 0.00002 with a cosine scheduler, use a gradient accumulation step size of 2, do not freeze any modules, and follow other settings from the LoRA fine-tuning configuration of Qwen-VL 2.5 Instruct in (Zheng et al., 2024b). We fine-tune Qwen-VL 2.5 Instruct (Team, 2025) using the SFT algorithm within the framework of (Zheng et al., 2024b) for 2600 steps on a single A800 GPU.

### A.8.2 TRAJECTORY DIFFUSION MODEL FINETUNING

Originally, the context length of the trajectory generation model (Zhong et al., 2023b) is set to 6. Since our framework takes a single-frame initial scene as input, we retrain the model to align with this setup. Following the configuration of (Zhong et al., 2023b), we introduce two key modifications: (1) the context length is set to 1, and (2) the motion restriction mask for static vehicles is removed to allow more vehicles to collide with the ego vehicle. The trajectory generation model is trained for 80,000 steps.

## A.9 VIDEO GENERATION SETTINGS

In UniMLVG (Chen et al., 2024b), textual descriptions derived from ground truth frames are used as text conditions for video generation. However, this limits the diversity of generated videos, as

Table 10: Performance of SparseDrive in Section 4.4

| Methods | Sample-level $CR$ (%) ↑ | | | | Scene-level $CR$ ↑ | | | | NC↓ | TTC↓ |
|---|---|---|---|---|---|---|---|---|---|---|
| | 1s | 2s | 3s | Avg. | 1s | 2s | 3s | Avg. | | |
| Origin | 0.38 | 0.38 | 0.50 | 0.42 | 0.05 | 0.05 | 0.07 | 0.05 | 0.98 | 0.96 |
| Collision stage only | 16.42 | 26.63 | 34.17 | 25.74 | 1.35 | 2.20 | 2.82 | 2.12 | 0.67 | 0.62 |
| Two-stage simulation | 7.57 | 15.23 | 22.32 | 15.04 | 0.96 | 1.94 | 2.84 | 1.91 | 0.74 | 0.67 |

Table 11: Performance of DiffusionDrive in Section 4.4

| Methods | Sample-level $CR$ (%) ↑ | | | | Scene-level $CR$ ↑ | | | | NC↓ | TTC↓ |
|---|---|---|---|---|---|---|---|---|---|---|
| | 1s | 2s | 3s | Avg. | 1s | 2s | 3s | Avg. | | |
| Origin | 0.19 | 0.24 | 0.44 | 0.29 | 0.02 | 0.03 | 0.06 | 0.04 | 0.98 | 0.97 |
| Collision stage only | 15.83 | 26.70 | 34.52 | 25.68 | 1.30 | 2.20 | 2.85 | 2.12 | 0.67 | 0.62 |
| Two-stage simulation | 10.63 | 18.68 | 25.17 | 18.16 | 1.35 | 2.38 | 3.21 | 2.31 | 0.74 | 0.66 |

discrepancies between the vehicle trajectories in the generated video and those in the ground truth can lead to inaccuracies in object and background descriptions. Therefore, we only use time and weather descriptions in text condition, preserving the controllability of modifying temporal and weather attributes while evading textual inconsistencies when vehicle trajectories are altered. Subsequently, multi-view images from the initial scene are used as reference frames for the first roll-out. We convert the full-scene trajectory information into 3D bounding boxes and map lanes in the camera coordinate system and explicit viewpoint modeling, which are used as conditioning information to generate views that accurately follow the specified trajectories. Each roll-out produces 19 frames at 12 Hz. The last generated frame for each generation is then used as the reference frame for the next roll-out, and through autoregressive generation, the entire safety-critical trajectory is rendered into a multi-view video from the ego-vehicle's perspective.

## A.10 EFFICIENCY AND COST ANALYSIS

**Training**: We use GRPO to train the VLM, which consumes 80 GB of GPU memory. The training runs on four GPUs for 88 hours, resulting in 352 GPU hours. To align with our trajectory simulation from single-image initial scenes, we also train a one-step context trajectory diffusion model, which uses 8 GB of memory and trains for 15 hours on a single GPU.

**Inference**: The VLM infers in ~22 seconds with 28 GB memory. Each trajectory simulation stage takes ~2 minutes (<3 GB), totaling ~4 minutes. Video generation requires ~21 minutes (9-second video) and uses 40 GB, making it the main bottleneck. In total, generating one video without failure takes approximately 25 minutes. Considering the time overhead resulting from selecting inappropriate adversarial vehicles and failures in the evasion stage, the average time to generate a complete video increases to approximately 36 minutes, which is comparable with current driving video generators. We plan to accelerate the video synthesis process using techniques such as distillation or by incorporating a 3D VAE in future work.

## A.11 CLOSED-LOOP METRICS SETTINGS

The NAVSIM (Dauner et al., 2024) benchmark evaluation uses non-reactive simulations and closed-loop metrics for a comprehensive assessment. We focus on two collision-related metrics: no collisions (NC) and time-to-collision (TTC). We make some adjustments to better evaluate safety-critical data generation and to align with our dataset. Since our dataset annotations use 12 Hz interpolation (Wang et al., 2022), we set the closed-loop frequency to 12 Hz to ensure consistency with the annotations. By default, UniAD predicts future trajectories at 2 Hz over a a 3-second horizon, so we also evaluate TTC and NC on 3-second trajectory horizon. Both NC and TTC are set to the default value of 1. If the ego vehicle collides with another vehicle within 3 seconds, NC is set to 0. TTC is set to 0 if, for

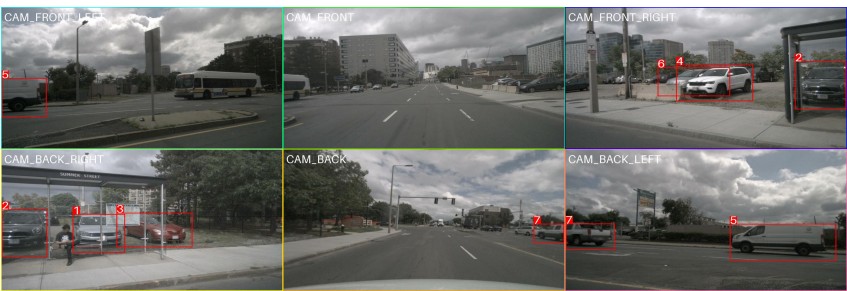

Figure 8: In this example, the nearest-vehicle rule would select Vehicle 2, which is blocked by an obstacle and therefore cannot produce a physically feasible scenario, whereas the VLM determines that there is no suitable vehicle in the scene.

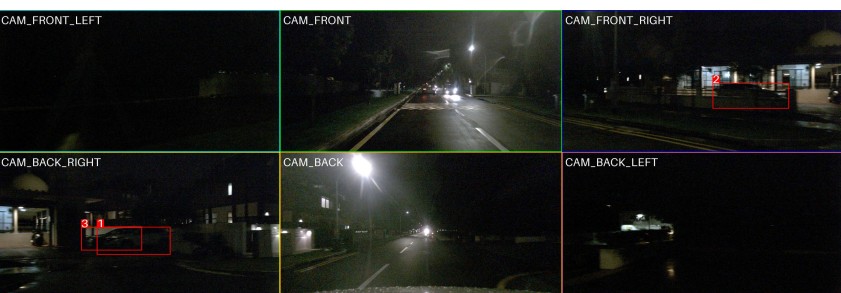

Figure 9: In this example, the nearest-vehicle rule would select Vehicle 2, which is blocked by an obstacle and therefore cannot produce a physically feasible scenario, whereas the VLM determines that there is no suitable vehicle in the scene.

any simulation step within the 4s horizon, the ego-vehicle's time-to-collison, when projected forward with a constant velocity and heading, is less than 1 second. Similarly to the collision rate, we make one adaptation: if a collision already occurs at the initial step, we consider such samples as invalid initializations and exclude them from our evaluation.

## A.12 BASELINE SELECTOR SETTINGS

**Random-neighbor rule:** Candidate vehicles are selected as adversaries based on their lane positions relative to the ego vehicle. Specifically, vehicles located within a predefined lane proximity (we select one lane) as the ego are identified, and one of them is randomly chosen as the adversary.

**Nearest-neighbor rule:** The adversarial vehicle is chosen as the one that is spatially closest to the ego vehicle in the initial frame (at 0 s). This rule emphasizes simplicity by selecting the nearest neighbor at the start of the scenario, regardless of subsequent speed, orientation, or lane constraints.

**Precision-first rule:** Candidate vehicles are first identified based on the following criteria; among them, the nearest one is selected.

- Both the ego vehicle and the candidate have current speeds greater than 2 m/s.
- At both 0 s and 2 s, the distance between the ego vehicle and the candidate is within the range of 10–30 m.
- At both 0 s and 2 s, the orientation difference between the ego vehicle and the candidate is less than 108 degrees.

## A.13 GENERATED VIDEOS USED FOR EVALUATION.

The videos used in our evaluation are generated under a fixed set of 250 samples, randomly selected from the val split, hereafter referred to as the base dataset. The following sections provide the video generation process in each experiment.

Prompt: You are a collision scenario analysis expert. Based on the traffic scenario described in the input images, your task is to identify the vehicle most likely to generate collision with the ego vehicle. The scene consists of six camera views surrounding the ego vehicle, arranged as follows: The first row includes three images: FRONT LEFT, FRONT, and FRONT RIGHT. The second row includes three images: BACK RIGHT, BACK, and BACK LEFT. Potential Dangerous Vehicles are highlighted with red boxes, and each vehicle's ID is labeled in the top-left corner of the respective box. Select the one most likely to have its future trajectory modified (through manual intervention) to produce the collision with the ego vehicle. The speed of any car other than ego vehicle can be adjusted, as long as it is in accordance with the laws of physics, so there is no need to analyze the speed of other cars. If no vehicle is suitable for this task, please respond that 'no vehicle is appropriate'. In the current scenario, the initial speed of the ego vehicle is $v$ m/s.

Figure 10: Original Prompt used in our VLM-based selection.

**Videos used in Section 4.2**: In this section, we compare the generated videos under three baselines: Origin, Naive, and our proposed SafeMVDrive. For the SafeMVDrive set, we apply our full framework to the base dataset and ultimately obtain 41 collision-evasion videos. For the Origin set, we start from the same 41 initial scenarios used in the SafeMVDrive set and convert their original NuScenes trajectories into videos. For the Naive set, we apply the naive baseline to all 250 initial scenarios in the base dataset and obtain 72 valid collision trajectories, which are then converted into videos. We evaluate the collision rates of the planner using videos on these three sets. Since FID scores are empirically affected by the number of images used in evaluation—more images generally lead to lower FID values—for fairness, we randomly sample 41 videos from the Naive set to compute FID. The videos used in the user study are described in Section A.2.

**Videos used in Section 4.4**: In this section, we compare the generated videos under three methods: Origin, Collision Stage Only, and Two-Stage Simulation. For the Two-Stage Simulation set, we apply our full framework to the base dataset and ultimately obtain 41 collision-evasion videos. For the Origin set, we start from the same 41 initial scenarios used in the Two-Stage Simulation set and convert their original NuScenes trajectories into videos. For the Collision Stage Only set, we start from the same 41 initial scenarios used in the Two-Stage Simulation set and skip the second simulation to generate videos and eventually get 41 collision videos. We evaluate the collision rates of the planner using videos and fid on these three sets. The videos used in the user study are described in Section A.2.

Table 12: Ablation Study on Loss Functions in the Two-Stage Evasion Trajectory Generator.

| CONFIGURATION | CSR ↑ | ESR ↑ | COLLISION RATE ↓ | OFF-ROAD RATE ↓ | REALISM ↓ | CLOSEST DISTANCE ↓ |
|---|---|---|---|---|---|---|
| WHOLE LOSSES | 0.750 | 0.402 | 0.042 | 0.002 | 0.312 | 5.37 |
| $-L_{adv}$ IN COLLISION STAGE | 0.471 | 0.703 | 0.034 | 0.000 | 0.308 | 9.11 |
| $-L_{no\_coll}$ IN COLLISION STAGE | 0.735 | 0.410 | 0.141 | 0.004 | 0.312 | 6.07 |
| $-L_{on\_road}$ IN COLLISION STAGE | 0.765 | 0.490 | 0.053 | 0.065 | 0.314 | 5.37 |
| $-L_{no\_coll}$ IN EVASION STAGE | 0.770 | 0.127 | 0.024 | 0.000 | 0.313 | 6.32 |
| $-L_{on\_road}$ IN EVASION STAGE | 0.770 | 0.304 | 0.057 | 0.007 | 0.310 | 5.23 |

A.14 ABLATION STUDY ON LOSS FUNCTIONS IN THE TWO-STAGE EVASION TRAJECTORY GENERATOR

We conduct ablation studies on the loss functions used in our two-stage evasion trajectory simulator. On 250 validation scenes, we first use the VLM selector to identify safety-critical candidates. Then, we remove one specific loss from the two-stage simulation while keeping the remaining loss terms unchanged to simulate.

We report the following metrics:

- **Collision Success Rate (CSR)**: the proportion of adversarial vehicles that successfully collide with the ego vehicle during collision simulation. A higher value is better.

- **Evasion Success Rate (ESR)**: the proportion of adversarial vehicles that successfully evade during evasion simulation. A higher value is better.

- **Collision Rate**: in the final trajectories, the proportion of adversarial vehicles that collide with any vehicle. Since these trajectories are later used for multi-view video simulation, and collision cases cannot be rendered, a lower value is preferred. This metric follows the implementation in CTG (Zhong et al., 2023b).

- **Off-Road Rate**: in the final trajectories, the proportion of adversarial vehicles that enter non-drivable areas. A lower value is better. This metric follows the implementation in CTG (Zhong et al., 2023b).

- **Realism**: in the final trajectories, the degree to which the trajectories resemble real-world behavior. In accordance with (Zhong et al., 2023b), we compare the statistical distribution between simulated trajectories and real-world trajectories. A lower value indicates better realism. This metric follows the implementation in CTG (Zhong et al., 2023b).

- **Closest Distance**: in the final trajectories, the minimum distance between the adversarial vehicle and the ego vehicle, measured by the distance between their center points, which reflects the potential danger level. A lower value is better.

The experimental results are shown in Table 12. The results demonstrate that each of our loss terms plays a crucial role. Removing $L_{adv}$ during the collision stage leads to a higher Closest Distance, indicating a lower safety criticality of the scenes. Removing $L_{no\_collision}$ results in a higher Collision Rate in the final trajectories. Removing $L_{on\_road}$ increases the Off-Road Rate in the final trajectories. During the evasion stage, removing $L_{no\_collision}$ decreases the Evasion Success Rate (ESR), resulting in fewer generated scenarios, while removing $L_{on\_road}$ similarly increases the Off-Road Rate in the final trajectories. These results verify the rationality and necessity of our loss design.

A.15 HYPERPARAMETERS STUDY OF THE LOSSES USED IN THE TWO-STAGE EVASION TRAJECTORY GENERATOR

In this section, we investigate the hyperparameters that control the contributions of different loss terms in the two-stage simulation. The positions of these hyperparameters can be found in Equations (4) and (5) in the main text. In addition to these, we also conduct hyperparameter studies on the weight decay rate factor $\lambda$. Similar to the previous section, we first use the VLM selector to identify safety-critical candidates on the 250 validation scenes. After that, we vary the hyperparameter corresponding to a specific loss term while keeping the other parameters fixed and then perform the two-stage simulation. We adopt the same evaluation metrics as in the previous section.

The experimental results are shown in Table 13, 15, 14, 16, 17, and 18. We vary the hyperparameters controlling the loss contributions with values $\{0, 1, 50\}$. Overall, setting the value to 0 generally leads to worse performance across various metrics, indicating the necessity of each individual loss term. On the other hand, when the value is within the range of 1 to 50, the differences among the metrics are relatively small, suggesting that our framework is not highly sensitive to hyperparameter selection.

For the weight decay factor $\lambda$, we evaluate settings of 0, 0.9, and 1. A value of 0 means that the loss is computed using only the prediction at timestamp 1, while a value of 1 averages the loss across all timestamps (i.e., no decay is applied). We observe that the best performance across all metrics is

Table 13: Ablation Study on $\alpha$ in Collision Stage.

| CONFIGURATION | CSR ↑ | ESR ↑ | COLLISION RATE ↓ | OFF-ROAD RATE ↓ | REALISM ↓ | CLOSEST DISTANCE ↓ |
|---|---|---|---|---|---|---|
| $\alpha = 0$ | 0.471 | 0.703 | 0.034 | 0.000 | 0.308 | 9.11 |
| $\alpha = 1$ (DEFAULT) | 0.750 | 0.402 | 0.042 | 0.002 | 0.312 | 5.37 |
| $\alpha = 50$ | 0.765 | 0.404 | 0.054 | 0.003 | 0.311 | 5.33 |

Table 14: Ablation Study on $\beta$ in Collision Stage.

| CONFIGURATION | CSR ↑ | ESR ↑ | COLLISION RATE ↓ | OFF-ROAD RATE ↓ | REALISM ↓ | CLOSEST DISTANCE ↓ |
|---|---|---|---|---|---|---|
| $\beta = 0$ | 0.735 | 0.410 | 0.141 | 0.004 | 0.312 | 6.07 |
| $\beta = 1$ | 0.735 | 0.440 | 0.083 | 0.005 | 0.311 | 5.63 |
| $\beta = 50$ (DEFAULT) | 0.750 | 0.402 | 0.042 | 0.002 | 0.312 | 5.37 |

achieved when $\lambda = 0.9$, which demonstrates the importance of applying a temporal weight decay in our loss design.

### A.16 PERFORMS ON DIFFERENT SENSOR CONFIGURATIONS

In this section, we perform experiments on the nuScenes dataset to assess how changes in camera parameters impact the generated video quality. We randomly perturb the extrinsic parameters of all cameras—up to 5 cm in translation and 2° in rotation along the x, y, and z axes. Since the initial camera images are captured using the original configuration and become misaligned after perturbation, we exclude them from the generation inputs. Other conditioning signals, such as bounding boxes and map information, are retained, and their projected positions in the camera views change accordingly as the extrinsic parameters vary. For comparison, we also generate videos under the original camera configuration without using the initial camera images as conditioning input. The results in Figure 19 show that FID, sample-level CR, and scene-level CR remain similar before and after perturbation, indicating that our framework demonstrates a degree of transferability to different camera parameters.

### A.17 LIMITATIONS

Since this is the first work to generate multi-view safety-critical driving videos in the real-world domain, we have several limitations. One is the reliance on the complete initial scene configuration, which restricts its ability to generate scenarios directly from raw multi-view camera inputs. Additionally, although our framework uses guidance signals to generate annotations, it lacks a mechanism to discard outdated or irrelevant ones—e.g., vehicles that have exited the ego's view. While this does not affect the evaluation of planning-related metrics, it may have some impact on perception-related evaluation metrics. Consequently, our dataset cannot be directly applied to tasks such as 3D detection or BEV segmentation for training and evaluation. Future research could address these challenges by reducing dependency on dense annotations and incorporating dynamic filtering strategies to maintain temporal relevance in the guidance signals.

### A.18 THE USE OF LARGE LANGUAGE MODELS

In this work, large language models (LLMs) are employed primarily for polishing purposes. Specifically, they are used to improve the clarity, fluency, and readability of the manuscript without altering the underlying technical content. The use of LLMs is restricted to language refinement, such as correcting grammatical errors, enhancing sentence structure, and improving overall coherence. Importantly, no novel ideas, experimental results, or conceptual contributions are generated by the LLMs; all scientific content and findings presented in this paper are entirely the work of the authors.

Table 15: Ablation Study on $\gamma$ in Collision Stage.

| CONFIGURATION | CSR ↑ | ESR ↑ | COLLISION RATE ↓ | OFF-ROAD RATE ↓ | REALISM ↓ | CLOSEST DISTANCE ↓ |
|---|---|---|---|---|---|---|
| $\gamma = 0$ | 0.765 | 0.490 | 0.053 | 0.065 | 0.314 | 5.37 |
| $\gamma = 1$ (DEFAULT) | 0.750 | 0.402 | 0.042 | 0.002 | 0.312 | 5.37 |
| $\gamma = 50$ | 0.779 | 0.396 | 0.059 | 0.003 | 0.311 | 5.60 |

Table 16: Ablation Study on $\beta$ in Evasion Stage.

| CONFIGURATION | CSR ↑ | ESR ↑ | COLLISION RATE ↓ | OFF-ROAD RATE ↓ | REALISM ↓ | CLOSEST DISTANCE ↓ |
|---|---|---|---|---|---|---|
| $\beta = 0$ | 0.750 | 0.127 | 0.024 | 0.000 | 0.313 | 6.32 |
| $\beta = 1$ (DEFAULT) | 0.750 | 0.402 | 0.042 | 0.002 | 0.312 | 5.37 |
| $\beta = 50$ | 0.750 | 0.422 | 0.041 | 0.003 | 0.311 | 4.92 |

Table 17: Ablation Study on $\gamma$ in Evasion Stage.

| CONFIGURATION | CSR ↑ | ESR ↑ | COLLISION RATE ↓ | OFF-ROAD RATE ↓ | REALISM ↓ | CLOSEST DISTANCE ↓ |
|---|---|---|---|---|---|---|
| $\gamma = 0$ | 0.750 | 0.304 | 0.057 | 0.007 | 0.310 | 5.23 |
| $\gamma = 1$ (DEFAULT) | 0.750 | 0.402 | 0.042 | 0.002 | 0.312 | 5.37 |
| $\gamma = 50$ | 0.750 | 0.363 | 0.002 | 0.048 | 0.310 | 5.59 |

Table 18: Ablation Study on $\lambda$.

| CONFIGURATION | CSR ↑ | ESR ↑ | COLLISION RATE ↓ | OFF-ROAD RATE ↓ | REALISM ↓ | CLOSEST DISTANCE ↓ |
|---|---|---|---|---|---|---|
| $\lambda = 0$ | 0.640 | 0.011 | 0.182 | 0.091 | 0.336 | 8.61 |
| $\lambda = 0.9$ (DEFAULT) | 0.750 | 0.402 | 0.042 | 0.002 | 0.312 | 5.37 |
| $\lambda = 1$ | 0.765 | 0.433 | 0.060 | 0.004 | 0.317 | 5.85 |

Table 19: Performance under unseen camera configurations.

| METHODS | SAMPLE-LEVEL $CR$ (%) ↑ | | | | SCENE-LEVEL $CR$ ↑ | | | | FID ↓ |
|---|---|---|---|---|---|---|---|---|---|
| | 1s | 2s | 3s | AVG. | 1s | 2s | 3s | AVG. | |
| ORIGINAL CAMERA | 7.58 | 18.96 | 28.09 | 18.21 | 1.32 | 3.32 | 4.90 | 3.18 | 31.70 |
| MODIFIED CAMERA | 7.58 | 19.24 | 28.51 | 18.44 | 1.34 | 3.39 | 4.98 | 3.24 | 35.85 |

