# OpenReview forum: "SafeMVDrive: Multi-view Safety-Critical Driving Video Generation in the Real World Domain"
_ICLR.cc/2026/Conference — ICLR 2026 Conference Withdrawn Submission_

### Official Review · Reviewer_c85w · 2025-10-20

**Soundness:** 3
**Presentation:** 3
**Contribution:** 2
**Rating:** 6
**Confidence:** 4

**Summary:**

This paper addresses multi-view, safety-critical video generation in the driving domain. It introduces SafeMVDrive, a framework that integrates a safety-critical trajectory simulator with a multi-view video generator and leverages visual context via a fine-tuned vision–language model (VLM) to select safety-critical vehicles. Experiments demonstrate that SafeMVDrive generates high-quality multi-view driving videos and induces 30% more collisions than the original NuScenes data, highlighting its effectiveness for safety-critical scenario synthesis.

**Strengths:**

1. The paper is clearly written and presents rich, detailed content.
2. The ability to generate safety-critical driving scenarios addresses an important real-world need, as prior approaches have struggled to produce such rare but crucial events.
3. The paper provides extensive and clear visualizations, which effectively illustrate the high-quality simulation performance of the proposed method.

**Weaknesses:**

1. The paper demonstrates notable practical and engineering value; however, its technical novelty is somewhat limited. The approach primarily integrates existing components—multi-view conditional generators, trajectory proposal modules, and VLM-based selection.
2. The academic value is difficult to assess, as it lacks comparisons with related methods (though such methods may themselves be scarce) and "safety" is inherently a broad concept. Overall, its engineering value outweighs its academic value.
3. The term safety-critical is rather broad, and the paper lacks a precise definition. Moreover, the current work addresses only safety issues arising from other vehicles’ trajectories, whereas real-world safety concerns extend beyond this scope. It is recommended that the paper provide a more concrete and detailed formulation of the problem.
4. How well does the VLM-based selection generalization, and what happens if the selection fails? Are the final safety-critical scenarios manually adjusted in such cases? My understanding is that the method primarily leverages the VLM to facilitate the generation of more safety-critical simulation scenarios—is this correct?

**Questions:**

1. Does training a policy model on generated safety-critical scenarios enhance its robustness to dangerous driving situations?
2. Since interactive cases in NuScenes are relatively rare, how is the ground truth for safety-critical vehicles selected when using the VLM?
3. Metrics relying on a single planner tend to be susceptible to fluctuations. It is therefore recommended to evaluate a broader range of planning methods to verify the consistency of results.
4. What are the practical potential application scenarios of this pipeline? Is it intended to serve as a corner case evaluation benchmarks or to train more robust planners?

---

> ### Author Response · Authors · 2025-11-21
>
> We sincerely appreciate your valuable time and comments. We have uploaded the rebuttal version and marked all the changes in blue.
>
> **W1: Notable engineering value but technical novelty limited. Just intergrates existing three component.**
>
> Thank you very much for recognizing the notable practical and engineering value of our work. We highlight that our technical novelty lies in strategically combining and **adapting** existing tools to achieve **the first framework to generate high-quality, multi-view, safety-critical videos in real world domains**, rather than simply combining multiple existing ideas.
>
> **Specifically, we make two key innovations:**
>
> **(1)** To select vehicles that can generate physically plausible collision scenarios, we introduce a novel approach that leverages visual information via a Vision-Language Model (VLM). We propose an automated annotation pipeline to construct a dataset and fine-tune the VLM for this task. It addresses the limitations of previous rule-based methods that fail to capture complex driving scenarios and often lead to physically infeasible selections.
>
> **(2)** To generate high-quality safety-critical videos, we adapt existing collision trajectory generation into a two-stage collision–evasion trajectory framework. This addresses the issue that directly coupling a collision-trajectory simulator with a video generator would substantially degrade the realism of the generated videos.
>
> Overall, these contributions work together to produce realistic, diverse, and safety-critical scenarios that meaningfully support both the evaluation and improvement of end-to-end planners.
>
> **W2: Academic value is difficult to assess as lack comparison and "safety" is inherently a broad concept.**
>
> Thank you for the thoughtful comment and for acknowledging the engineering value of our work. We believe that academic value arises from two types of contributions: (i) tackling an important problem for which no prior solutions existed, and (ii) demonstrating significant improvements over well-motivated baselines. Our work is of the first type—it provides the first framework capable of generating multi-view, safety-critical videos grounded in real-world data.
>
> Because no prior method addresses this problem, direct comparisons are not feasible. To provide a meaningful point of reference, we constructed a naive baseline by pairing an existing safety trajectory generator with a video synthesis model. Our method achieves markedly higher video realism and scenario fidelity than this baseline. We expect that the research community will build upon this direction, enabling more extensive comparative studies in the future.
>
> With respect to the notion of safety, we agree that *safety* can be broad in general discussions. In our study, however, we adopt a precise and operational definition focused on a specific class of high-risk events: collisions between the ego vehicle and another vehicle. These events represent some of the most consequential hazards for autonomous driving systems. Under this definition, safety is directly measurable through the planner’s collision rate. Lower collision rates indicate less safety-critical scenarios, whereas higher rates correspond to more challenging and safety-critical ones.
>
> **W3: The term safety-critical is rather broad, and the paper lacks a precise definition.**
>
> As noted above, *safety-critical* is indeed a broad term. In this paper, we adopt a precise definition focused on a specific and well-scoped class of events: scenarios involving potential collisions between the ego vehicle and an adversarial vehicle. We have clarified this definition in the first paragraph of the Related Work section to ensure that the problem formulation is explicit and unambiguous.
>
> We agree that real-world safety encompasses a wider range of interactions. In future work, we plan to extend our framework to generate additional types of safety-critical scenarios, including collisions between non-ego vehicles and interactions involving vulnerable road users such as pedestrians. We provide further discussion of these broader scenario classes in Appendix A.1.

---

> ### Author Response · Authors · 2025-11-21
>
> **W4: How well the VLM-based selection generalizes, whether manual adjustment is needed when the selection fails, and what the role of the VLM is.**
>
> We evaluate the current VLM on both the training set and the held-out test set, and the results in below table show that its selection behavior generalizes well.
>
> |  | Precision | Recall | F1-score |
> | --- | --- | --- | --- |
> | Train | 0.858 | 0.812 | 0.8344 |
> | Val | 0.750 | 0.675 | 0.710
>  |
>
> If the VLM makes an incorrect selection or the second-stage evasion simulation fails, this **does not**  need to be manually adjusted. Our two-stage simulation pipeline contains explicit filtering: adversarial vehicles chosen based on incorrect VLM estimations are discarded if they do not lead to effective collisions with the ego vehicle in the collision-trajectory simulation (e.g., they collide with other vehicles first or leave the drivable area). Similarly, cases where the ego vehicle cannot successfully execute an evasion trajectory are also filtered out. As a result, we do not need to manually adjust scenarios.
>
> Our main goal of using VLM is to ensure that the selected safety-critical vehicles are physically feasible.  As shown in Table 2, the VLM-based method achieves the highest selection F1 score, and as shown in Appendix A.6, when obvious obstacles are present, rule-based methods may incorrectly select blocked vehicles, whereas the VLM successfully avoids these vehicles and selects the correct one.
>
> **Q1:  Does training a policy model on generated safety-critical scenarios enhance its robustness?**
>
> Thank you very much for your valuable suggestion. We conduct an experiment to verify the usefulness of the videos generated by SafeMVDrive for improving an end-to-end autonomous driving planner in terms of collision avoidance. Specifically, we split the generated data into training and validation sets with a ratio of roughly 4:1, and mix the training set with the original NuScenes training dataset to fine-tune the light weight E2E planner DiffusionDrive (2025 CVPR spotlight) [1] for 10 epochs. We then evaluate the model on both the NuScenes validation set and our generated SafeMVDrive validation set. From the results below, we can see that after training the end-to-end planner, collision-related metrics decrease significantly on the SafeMVDrive validation dataset, and also decrease on NuScenes validation dataset. This further demonstrates the effectiveness of our framework. The corresponding experimental results have been added to Section 4.5 of the main paper.
>
>
> | Evaluation Set | Model |  | Sample level CR(%) **↓** |  |  |  | Scene level CR **↓** |  |  | NC **↑** | TTC **↑** |
> | :---: | :---: | :---: | :---: | :---: | :---: | :---: | :---: | :---: | :---: | :---: | :---: |
> |  |  | 1 | 2 | 3 | averaged | 1 | 2 | 3 | averaged |  |  |
> | NuScenes Val | Base | 0.11 | 0.15 | 0.28 | 0.18 | 0.03 | 0.05 | 0.08 | 0.05 | 0.992 | 0.976 |
> |   | Finetuned | 0.06 | 0.11 | 0.21 | 0.13 | 0.02 | 0.03 | 0.06 | 0.04 | 0.993 | 0.979 |
> | SafeMVdrive Val | Base | 6.63 | 17.6 | 25.68 | 16.64 | 0.81 | 2.16 | 3.15 | 2.04 | 0.743 | 0.684 |
> |  | Finetuned | 3.06 | 7.4 | 12.59 | 7.68 | 0.37 | 0.91 | 1.54 | 0.94 | 0.822 | 0.763 |
>
>
> [1] Liao, B., Chen, S., Yin, H., Jiang, B., Wang, C., Yan, S., ... & Wang, X. (2025). Diffusiondrive: Truncated diffusion model for end-to-end autonomous driving. In Proceedings of the Computer Vision and Pattern Recognition Conference (pp. 12037-12047).
>
> **Q2: How is the ground truth for safety-critical vehicles selected when using the VLM?**
>
> To generate fine-tuning labels for the VLM-GRPO, we randomly select samples from nuScence training set and exhaustively traverse all vehicles in the initial scene (excluding those that are too distant) as potential adversarial vehicles. For each candidate, we employ the collision simulation phase of our two-stage simulator to determine feasibility: if a candidate vehicle collides with the ego vehicle before colliding with any other vehicles or entering a non-drivable area, we regard it as a valid adversarial vehicle and include it as part of the ground truth.

---

> ### Author Response · Authors · 2025-11-21
>
> **Q3:  Evaluate a broader range of planning methods to verify the consistency of results.**
>
> Thank you very much for your suggestions. We have added two additional end-to-end planners, SparseDrive [1] and DiffusionDrive [2], to test on the videos we generated. We replace Table 1 (evaluation of different baselines) and Table 4 (ablation study of the two-stage simulation)  in the main paper with the average results. The experimental results are similar: our SafeMVDrive-generated videos pose significant challenges to the planners compared to those generated with the original trajectories, resulting in a substantial improvement in collision-related metrics. This demonstrates that the videos generated by our framework exhibit strong safety-critical characteristics.
>
>
> | Averaged Table 1 |  | Sample level CR（%） |  |  |  | Scene level  CR |  |  | NC | TTC |
> | :---: | :---: | :---: | :---: | :---: | :---: | :---: | :---: | :---: | :---: | :---: |
> |  | 1 | 2 | 3 | averaged | 1 | 2 | 3 | averaged |  |  |
> | Origin  | 0.57 | 0.9 | 1.64 | 1.04 | 0.07 | 0.12 | 0.21 | 0.13 | 0.98 | 0.96 |
> | Naive  | 18.65 | 34.04 | 46.4 | 33.03 | 1.46 | 2.66 | 3.62 | 2.58 | 0.37 | 0.28 |
> | SafeMVDrive | 6.77 | 15.25 | 23.67 | 15.23 | 0.86 | 1.94 | 3.02 | 1.94 | 0.77 | 0.69 |
>
>
>
> | Averaged Table 4 |  | Sample level CR（%） |  |  |  | Scene level CR |  |  | NC | TTC |
> | :---: | :---: | :---: | :---: | :---: | :---: | :---: | :---: | :---: | :---: | :---: |
> |  | 1 | 2 | 3 | averaged | 1 | 2 | 3 | averaged |  |  |
> | Origin | 0.57 | 0.9 | 1.64 | 1.04 | 0.07 | 0.12 | 0.21 | 0.13 | 0.98 | 0.96 |
> | Collision Stage Only | 17.53 | 31.15 | 38.98 | 29.22 | 1.46 | 2.59 | 3.24 | 2.43 | 0.67 | 0.63 |
> | Two-stage simulation  | 6.77 | 15.25 | 23.67 | 15.23 | 0.86 | 1.94 | 3.02 | 1.94 | 0.77 | 0.69 |
>
>
> [1] Sun, Wenchao, et al. "Sparsedrive: End-to-end autonomous driving via sparse scene representation." *2025 IEEE International Conference on Robotics and Automation (ICRA)*. IEEE, 2025.
>
> [2] Liao, B., Chen, S., Yin, H., Jiang, B., Wang, C., Yan, S., ... & Wang, X. (2025). Diffusiondrive: Truncated diffusion model for end-to-end autonomous driving. In Proceedings of the Computer Vision and Pattern Recognition Conference (pp. 12037-12047).
>
> **Q4: What are the practical potential application scenarios of this pipeline?**
>
> Our framework can generate a large number of safety-critical videos. So far, we have demonstrated two key applications: (1) construct datasets for open-loop evaluation of end-to-end planners in safety-critical scenarios (see answers to Q3 for results), and (2) generate training data to improve the collision-avoidance performance of end-to-end planners (see answers to Q1 for results).

---

> > ### Comment · Reviewer_c85w · 2025-11-27
> >
> > Thank you for the detailed response. The added experiments on policy robustness and broader planning methods strengthen the work’s validity and practical value. I am inclined to recommend acceptance of the paper.

---

> > > ### Author Response · Authors · 2025-11-27
> > >
> > > Thank you for taking the time to share your thoughtful comments and suggestions, and for your recognition — we deeply appreciate it.

---

### Official Review · Reviewer_ffnN · 2025-10-29

**Soundness:** 3
**Presentation:** 4
**Contribution:** 3
**Rating:** 8
**Confidence:** 4

**Summary:**

This paper proposes a method for synthesizing safety-critical scenario data, for which the authors design a VLM-based selector and a two-stage trajectory generator. By evaluating the quality of the generated videos and the validity rate of the trajectories, this paper demonstrates the effectiveness of the proposed pipeline. Long-tail scenarios are a critical challenge that autonomous driving must address. This paper represents a attempt to improve the performance of autonomous driving models on such long-tail scenarios.

**Strengths:**

One notable strength of this work is its clear articulation of a critical gap in existing generative models: despite recent advances in video synthesis, they fail to produce safety-critical driving scenarios—especially those involving potential collisions—that are vital for robustness evaluation of autonomous vehicles. The paper rightly positions this as a core challenge in handling long-tail events, which are rare yet consequential in real-world deployment.

This paper proposes a novel multi-stage framework: it first employs a VLM-based selector to identify a potentially colliding target vehicle, then uses a trajectory generator to sequentially produce a collision trajectory followed by an evasion trajectory, and finally leverages an existing video generation model—conditioned on the generated trajectories—to synthesize realistic driving videos.

**Weaknesses:**

The paper primarily evaluates two components: (1) the accuracy of the VLM-based adversarial vehicle selector, and (2) the performance of the synthesized data when used to test the UniAD planner, reporting metrics such as Collision Rate (CR), Near-Collision Rate (NC), and Time-to-Collision (TTC).
However, it does not address a crucial practical question: can the generated safety-critical scenarios be effectively used for training autonomous driving models? Demonstrating utility in evaluation is valuable, but the potential of the synthetic data to improve model robustness or safety during training—a primary goal of data synthesis—remains unexplored and should be discussed.

Furthermore, the authors directly adopt the pre-trained UniM-LVG video generator without fine-tuning. Given that UniM-LVG’s training data likely underrepresents safety-critical events (e.g., collisions or near-misses), its capacity to generate high-quality, physically plausible safety-critical videos is uncertain. The paper lacks empirical validation—such as visual fidelity checks, physics-based consistency analysis, or user studies—to confirm that the generated scenes are realistic and meaningful in these extreme cases.

**Questions:**

1. Please supplement the experiments with discussion or validation demonstrating that UniM-LVG can generate high-quality data for the safety-critical scenarios described in this paper.

2. Collisions represent only one type of safety-critical scenario. The paper should also discuss the cost and methodology required to extend the proposed approach to other types of safety-critical situations.

---

> ### Author Response · Authors · 2025-11-21
>
> We sincerely appreciate your valuable time and comments. We have uploaded the rebuttal version and marked all the changes in blue.
>
> **W1: Enhancement for End-to-end planner.**
>
> Thank you very much for your valuable suggestion. We conduct an experiment to verify the usefulness of the videos generated by SafeMVDrive for improving an end-to-end autonomous driving planner in terms of collision avoidance. Specifically, we split the generated data into training and validation sets with a ratio of roughly 4:1, and mix the training set with the original NuScenes training dataset to fine-tune the light weight E2E planner DiffusionDrive (2025 CVPR spotlight) [1] for 10 epochs. We then evaluate the model on both the NuScenes validation set and our generated SafeMVDrive validation set. From the results below, we can see that after training the end-to-end planner, collision-related metrics decrease significantly on the SafeMVDrive validation dataset, and also decrease on NuScenes validation dataset. This further demonstrates the effectiveness of our framework. The corresponding experimental results have been added to Section 4.5 of the main paper.
>
> | Evaluation Set | Model |  | Sample level CR(%) **↓** |  |  |  | Scene level CR **↓** |  |  | NC **↑** | TTC **↑** |
> | :---: | :---: | :---: | :---: | :---: | :---: | :---: | :---: | :---: | :---: | :---: | :---: |
> |  |  | 1 | 2 | 3 | averaged | 1 | 2 | 3 | averaged |  |  |
> | NuScenes Val | Base | 0.11 | 0.15 | 0.28 | 0.18 | 0.03 | 0.05 | 0.08 | 0.05 | 0.992 | 0.976 |
> |   | Finetuned | 0.06 | 0.11 | 0.21 | 0.13 | 0.02 | 0.03 | 0.06 | 0.04 | 0.993 | 0.979 |
> | SafeMVdrive Val | Base | 6.63 | 17.6 | 25.68 | 16.64 | 0.81 | 2.16 | 3.15 | 2.04 | 0.743 | 0.684 |
> |  | Finetuned | 3.06 | 7.4 | 12.59 | 7.68 | 0.37 | 0.91 | 1.54 | 0.94 | 0.822 | 0.763 |
>
>
> [1] Liao, B., Chen, S., Yin, H., Jiang, B., Wang, C., Yan, S., ... & Wang, X. (2025). Diffusiondrive: Truncated diffusion model for end-to-end autonomous driving. In Proceedings of the Computer Vision and Pattern Recognition Conference (pp. 12037-12047).
>
> **W2: The realism of the generated videos.**
>
> To demonstrate the realism of the generated videos, we report both the visual fidelity (e.g., FID score) and the human-evaluated natural score in Tables 1 and 4. The results show that the SafeMVDrive videos achieve scores that are very close to the Origin videos (approximately 90%).
>
> We further include an additional experiment to evaluate the **consistency** of the generated videos. We consider that stable tracking reflects physical plausibility, as tracking often fails when an object deforms unrealistically. Since our scenarios focus on collision avoidance between the ego vehicle and an adversarial vehicle, the adversarial vehicle is most likely to exhibit such inconsistencies. We therefore use StreamPETR [2] to track the adversarial vehicle in the generated videos and define *Subject Consistency* as the ratio between its tracked frame length and its actual frame length in the scene. As shown in the table below, the *Subject Consistency* of SafeMVDrive videos is very close to that of the Origin videos, whereas the consistency scores of the Naive baseline (where collision happens) drop substantially.
>
> Combining the results from the physics-based consistency analysis (Subject Consistency), user studies (Natural Score), and visual fidelity checks (FID), we conclude that the generated safety-critical scenario videos exhibit high realism. Thanks for your suggestion and we have added the new metric to the main paper.
>
>
>
>
> |  | Subject Consistency | Human Study (Natural Score) | FiD |
> | :---: | :---: | :---: | :---: |
> | Origin | 0.881  | 0.63 | 16.25 |
> | Naive | 0.659 | 0.21 | 23.35 |
> | SafeMVDrive | 0.859 | 0.56 | 20.63 |
>
>
> [1]: Liao, Bencheng, et al. "Diffusiondrive: Truncated diffusion model for end-to-end autonomous driving." *Proceedings of the Computer Vision and Pattern Recognition Conference*. 2025.
>
> [2]: Wang, Shihao, et al. "Exploring object-centric temporal modeling for efficient multi-view 3d object detection." *Proceedings of the IEEE/CVF international conference on computer vision*. 2023.

---

> ### Author Response · Authors · 2025-11-21
>
> **Q1: Supplement the experiments with discussion demonstrating that UniMLVG can generate high-quality safety-critical data.**
>
> UniMLVG is trained on 1,498 hours of diverse driving data (including OpenDV-Youtube, nuScenes, Waymo, and Argoverse2), providing strong out-of-distribution generalization that enables it to accurately represent our safety-critical collision-avoidance scenarios. In addition, its multi-task training scheme effectively reduces autoregressive error accumulation in long-horizon synthesis, which is essential since safety-critical events typically unfold over longer time spans.
>
> Based on these properties, we adopt UniMLVG as the backbone for generating the safety-critical video scenarios required by our framework. In the above answer of Weakness 2, we further validate that the resulting video quality is sufficient for our purposes using a new Subject Consistency score, as well as prior human-study naturalness scores and FID scores. We have updated Section 3.4 to include a more detailed explanation of how UniMLVG supports high-quality video generation in our framework. Thanks for your suggestion.
>
> **Q2: Discussion of the cost and methods for extending to other safety-critical scenarios.**
>
> Thank you for this insightful suggestion. We appreciate your emphasis on recognizing collision as a key safety-critical case. Indeed, constructing datasets of inter-vehicle collision scenarios has been an important research topic [1,2,3]. Along this line, our work is the first to generate real-domain, multi-view scenarios involving collision avoidance between the ego vehicle and another vehicle.
>
> We fully agree that generating a broader variety of safety-critical scenarios would be even more beneficial for autonomous driving. Our current framework focuses on safety-critical collision-avoidance scenarios between the ego vehicle and an adversarial vehicle. Since safety-critical situations are highly diverse, we next discuss how our approach could be extended to two additional common types of scenarios. (1): for collisions avoidance between two non-ego vehicles, one only need to replace the loss-guidance target for the adversarial vehicle with the other vehicle in the collision stage simulation.  (2): for safety-critical interactions between pedestrians and vehicles, one need to train a diffusion-based trajectory generator specifically for pedestrians and control their motion using a test-time loss-guidance strategy similar to that proposed in our paper. Furthermore, existing autonomous driving video generators still exhibit inadequate performance when conditioned on out-of-distribution pedestrian trajectories, indicating that stronger modeling of pedestrians is required to ensure high-quality generation in such scenarios.
>
> Thank you for your valuable suggestions. We have added the relevant content to Appendix A.1.
>
> [1]: Wang, Tianqi, et al. "Deepaccident: A motion and accident prediction benchmark for v2x autonomous driving." *Proceedings of the AAAI Conference on Artificial Intelligence*. Vol. 38. No. 6. 2024.
>
> [2]: Chang, Wei-Jer, et al. "Safe-sim: Safety-critical closed-loop traffic simulation with diffusion-controllable adversaries." *European Conference on Computer Vision*. Cham: Springer Nature Switzerland, 2024.
>
> [3]: Wang, Jingkang, et al. "Advsim: Generating safety-critical scenarios for self-driving vehicles." *Proceedings of the IEEE/CVF Conference on Computer Vision and Pattern Recognition*

---

> > ### Comment · Reviewer_ffnN · 2025-11-25
> >
> > Thank you for your reply. It has addressed my concerns. I’d like to retain my original score.

---

> > > ### Author Response · Authors · 2025-11-25
> > >
> > > Thank you again for taking the time to share your detailed comments and suggestions. We truly appreciate it.

---

### Official Review · Reviewer_GrpU · 2025-10-30

**Soundness:** 2
**Presentation:** 3
**Contribution:** 3
**Rating:** 6
**Confidence:** 4

**Summary:**

This works main contribution is to propose using VLM as an adversarial vehicle selector for multi-view video generation, which start with VLM selection and use trajecotry diffusion models for adversarial scenario genreation, then the genreated multiview videos are generated for E2E planner evaluation. Unlike most works that synthesize vehicle trajecotires, this work starts with visual cues from multi-view videos. The main limitation is that the generated videos are evaluated in an open-loop setting, where the E2E planner’s behavior may diverge from the input videos, limiting the realism and closed-loop consistency of the evaluation.

**Strengths:**

- This works focus on an important problem: Generating Safety-critical Multiview videos, especially more research  industry has now turned into End-to-End paradigm.
- Using Video to select Critical Object is an interesting direction, but it also potentially neglect safety-critical scenarios that are not in the perceptive field (collision or far-away vehicles)

**Weaknesses:**

- The main limitation and weakness is that this work is not closed-loop:
The multi-view videos used as input to the evaluated planner may diverge from the planner’s actual behavior, limiting the realism and validity of the evaluation. The 2-stage evasion stage in table 4 refinement is not the actual evaluated planner's behavior.
- **Table 2 may not be a fair comparison.** The proposed annotation strategy only identifies vehicles that can collide with the ego vehicle, which may miss adversarial agents farther away. In contrast, a random lane-based neighbor selection could reveal a wider range of challenging interactions. Moreover, some nearest-vehicle collisions (as shown in the qualitative videos) appear uninteresting, such as simple rear-end cases.
- Why is the evaluation limited to 3 seconds in table 4, rather than testing longer horizons for more realistic interactions?
- Minor weakness: Fine Tuning VLM w/ CTG++ results, this works proposes to auto label feasible collision vehicles by checking if collisions can happen before off-road and collided with other vehicles, this would potentially scenarios due to CTG++’s performance issue.

**Questions:**

- Can the Proposed framework handle Closed-loop simulation? For example, can we have planning algorithms 1 and 2, where the final scenarios may be different depending on the interactions?
- How well does the VLM handle **multi-view visual inputs**? In particular, does the framework tend to select vehicles primarily from the **front and rear views**, or are there also cases where **side-view vehicles** are chosen as adversarial agents?
- Report the inference speed of the proposed Pipelines

---

> ### Author Response · Authors · 2025-11-21
>
> We sincerely appreciate your valuable time and comments. We have uploaded the rebuttal version and marked all the changes in blue.
>
> **W1: Not closed-loop and ego vehicle’s trajectory in videos is not the evaluated planner‘s behavior.**
>
> Thank you for your thoughtful feedback. We agree that closed-loop evaluation provides the most realistic assessment of planner performance because it incorporates the feedback effect of the planner’s actions. At the same time, open-loop evaluation remains a standard and widely adopted methodology in end-to-end planning research [1,2], largely due to its efficiency and its use of visually realistic real-world trajectories.
>
> In open-loop evaluation—the setting we adopt—the driving trajectory in the input video is not required to correspond to the actions the planner itself would take. This does not invalidate the evaluation. A robust end-to-end planner is expected to output safe, collision-free trajectories for any plausible driving input, irrespective of whether that input reflects its own behavior. The divergence between the driving trajectory in the input video and the planner’s hypothetical closed-loop behavior does not diminish the validity of this safety assessment.
>
> We fully agree that closed-loop evaluation with world models is an important direction for future research. However, building a stable closed-loop pipeline with current video generative models remains highly challenging. Closed-loop execution would require frequent trajectory updates and repeated self-regressive video generations, which accumulate significant cumulative errors and degrade video fidelity, ultimately confounding the planner evaluation itself.
>
> In summary, open-loop video generation and closed-loop planner evaluation represent two complementary but distinct research tracks. Our method is designed to support the open-loop track by producing realistic, safety-critical visual scenarios that are well-suited for stress-testing end-to-end planners. While closed-loop evaluation is indeed valuable, it involves a different set of technical assumptions and remains an open problem for world-model-based systems.
>
> [1] Liao, B., Chen, S., Yin, H., Jiang, B., Wang, C., Yan, S., ... & Wang, X. (2025). Diffusiondrive: Truncated diffusion model for end-to-end autonomous driving. In Proceedings of the Computer Vision and Pattern Recognition Conference (pp. 12037-12047).
>
> [2] Zheng, Y., Yang, P., Xing, Z., Zhang, Q., Zheng, Y., Gao, Y., ... & Zhao, D. (2025). World4Drive: End-to-End Autonomous Driving via Intention-aware Physical Latent World Model. In Proceedings of the IEEE/CVF International Conference on Computer Vision .

---

> ### Author Response · Authors · 2025-11-21
>
> **W2: Table 2 may not be a fair comparison and collision examples appear uninteresting.**
>
> Thank you very much for your question. The comparison in Table 2 is in fact fair. When preparing the groundtruth labels in Table 2, we follow the automatic annotation process and **do not filter out distant vehicles** (which is different from how we construct the VLM training labels); instead, we simulate all vehicles in each scene. A distant vehicle is regarded as a successful selection as long as it can produce a valid collision with the ego vehicle within the duration of the generated video (without colliding with other vehicles or entering non-drivable areas). We have added a clear clarification in Section 4.3—thank you for pointing it out.
>
> Regarding the showcased video examples, there are also other types of cases in our video examples ([https://iclr-1.github.io/SMD/#Video Gallery](https://iclr-1.github.io/SMD/#Video%20Gallery)). For instance, in the first example, the leading vehicle cuts in while the ego vehicle steers and overtakes. In the sixth example, the front adversarial vehicle suddenly slows down, and the ego vehicle also decelerates to evade it. These examples illustrate that the generated scenarios are not limited to straightforward rear-end collisions but also include richer, more complex behaviors.
>
> Beyond being interesting, **the more important point is that the generated videos should provide practical value for autonomous driving systems**. Our previous results (Table 1) show that using them to test end-to-end planners substantially increases collision rate. We further evaluate their value for training planner: we mix the generated data (4:1 train–test split) with NuScenes training set and finetune a lightweight planner DiffusionDrive (2025 CVPR spotlight) [1]  for 10 epochs. The trained E2E planner shows a clear reduction in collision-related metrics on the SafeMVDrive validation set, while have better performance on the original NuScenes validation set, further validating the effectiveness of our framework. The corresponding experimental results have been added to Section 4.5 of the main paper.
>
>
> | Evaluation Set | Model |  | Sample level CR(%) **↓** |  |  |  | Scene level CR **↓** |  |  | NC **↑** | TTC **↑** |
> | :---: | :---: | :---: | :---: | :---: | :---: | :---: | :---: | :---: | :---: | :---: | :---: |
> |  |  | 1 | 2 | 3 | averaged | 1 | 2 | 3 | averaged |  |  |
> | NuScenes Val | Base | 0.11 | 0.15 | 0.28 | 0.18 | 0.03 | 0.05 | 0.08 | 0.05 | 0.992 | 0.976 |
> |   | Finetuned | 0.06 | 0.11 | 0.21 | 0.13 | 0.02 | 0.03 | 0.06 | 0.04 | 0.993 | 0.979 |
> | SafeMVdrive Val | Base | 6.63 | 17.6 | 25.68 | 16.64 | 0.81 | 2.16 | 3.15 | 2.04 | 0.743 | 0.684 |
> |  | Finetuned | 3.06 | 7.4 | 12.59 | 7.68 | 0.37 | 0.91 | 1.54 | 0.94 | 0.822 | 0.763 |
>
>
> [1] Liao, B., Chen, S., Yin, H., Jiang, B., Wang, C., Yan, S., ... & Wang, X. (2025). Diffusiondrive: Truncated diffusion model for end-to-end autonomous driving. In Proceedings of the Computer Vision and Pattern Recognition Conference (pp. 12037-12047).
>
> **W3: Why is the evaluation limited to 3 seconds in Table 4.**
>
> In Tables 1 and 4, we follow the standard open-loop evaluation protocol. The planner predicts a 3-second future trajectory starting from each frame, and the collision rate, NC, and TTC are computed over this 3-second horizon. In other words, the evaluation is performed using a sliding-window scheme: each window covers a 3-second interval, and the window slides frame by frame over the entire 9-second generated video. The final results are obtained by averaging over all windows.
>
> **W4: Minor weakness: Auto-labeling for VLM fine-tuning may be affected by trajectory simulator’s performance.**
>
> Thank you for your valuable feedback. The CTG framework is a widely adopted controllable diffusion-based simulation framework, and many subsequent works have built upon it to generate safety-critical trajectories [1, 2]. For this reason, we choose it to effectively simulate and label feasible collision scenarios.
>
> We acknowledge that the quality of the automatically labeled scenarios is influenced by the underlying performance of CTG. In practice, we find that CTG produces sufficiently accurate trajectories for our labeling pipeline; however, we agree that further improvements are possible. As future work, we plan to explore more advanced or diverse generative frameworks to increase the robustness and precision of the collision-feasibility labeling process.
>
> [1] Zhong, Ziyuan, et al. "Language-guided traffic simulation via scene-level diffusion." Conference on robot learning. PMLR, 2023.
>
> [2] Chang, Wei-Jer, et al. "Safe-sim: Safety-critical closed-loop traffic simulation with diffusion-controllable adversaries." *European Conference on Computer Vision*. Cham: Springer Nature Switzerland, 2024.

---

> ### Author Response · Authors · 2025-11-21
>
> **Q1: Can the Proposed framework handle Closed-loop simulation?**
>
> Closed-loop simulation is indeed an important direction, but our framework is currently not designed for closed-loop simulation. Closed-loop simulation based on world models requires very frequent auto-regressive data generation, which often leads to significant cumulative errors, resulting in generated quality that is far inferior to open-loop video generation [1,2,3]. Since our primary goal is to generate high-quality safety-critical data, we opt to produce full-length video clips rather than operate in a closed-loop manner.
>
> [1]: Chen, Rui, et al. "Unimlvg: Unified framework for multi-view long video generation with comprehensive control capabilities for autonomous driving." *arXiv preprint arXiv:2412.04842* (2024).
>
> [2]: Yang, Xuemeng, et al. "Drivearena: A closed-loop generative simulation platform for autonomous driving." *Proceedings of the IEEE/CVF International Conference on Computer Vision*. 2025.
>
> [3]: Yan, Tianyi, et al. "Drivingsphere: Building a high-fidelity 4d world for closed-loop simulation." *Proceedings of the Computer Vision and Pattern Recognition Conference*. 2025.
>
> **Q2: How well does the VLM handle multi-view visual inputs.**
>
> Thank you for your insightful question. Our current VLM demonstrates partial but not yet complete capability in handling multi-view visual inputs. In practice, the model can reliably selects vehicles appearing in the front and rear views. When a vehicle is visible in both a side view and a front/rear view, the VLM is generally able to identify it consistently. However, in the present implementation, vehicles that appear exclusively in side-view cameras remain challenging for the model to select.
>
> **As this is the first work to use a VLM for selecting safety-critical vehicles, our primary objective has been to ensure that the VLM can select safety-critical vehicles that lead to physically feasible collisions**. As shown in Table 2, the VLM-based method achieves the highest selection F1 score. Furthermore, as shown in Appendix A.6, when obvious obstacles are present, rule-based methods may incorrectly select blocked vehicles, whereas the VLM successfully avoids these vehicles and selects the correct one.
>
> We agree that extending the VLM’s ability to reason over side-view–only vehicles is an important next step. Future work will focus on enhancing multi-view spatial understanding, potentially through improved cross-view feature alignment or multi-view pretraining strategies.
>
> **Q3: Report the inference speed of the proposed Pipelines.**
>
> We test the generation speed on a single A800 GPU. The VLM infers in ~22 seconds. Each trajectory simulation stage takes ~2 minutes, totaling ~4 minutes. Video generation requires ~21 minutes (9-second video), making it the main bottleneck. In total, generating one video without failure takes approximately 25 minutes. Considering the time overhead resulting from selecting inappropriate adversarial vehicles and failures in the evasion stage, the average time to generate a complete video increases to approximately 36 minutes, which is comparable with current driving video generators. We plan to accelerate the video synthesis process in future work. Relevant details have been added to Appendix A.10.

---

### Official Review · Reviewer_izyK · 2025-10-30

**Soundness:** 2
**Presentation:** 3
**Contribution:** 3
**Rating:** 4
**Confidence:** 3

**Summary:**

This paper addresses the critical challenge of generating realistic, safety-critical data for the evaluation of modern autonomous driving (AD) systems. The authors identify a key gap in existing data generation methods: while they can produce trajectories, simulations, or single-view videos, they fail to generate the realistic multi-view video feeds that contemporary end-to-end AD systems actually consume. To solve this, the authors propose SafeMVDrive, a novel framework designed to be the first to synthesize multi-view, safety-critical driving videos in the real-world domain.
The core of their method involves coupling a safety-critical trajectory engine with a diffusion-based video generator, guided by three main contributions. First, they employ a fine-tuned Vision-Language Model (VLM) as an intelligent "adversary" to select the vehicle most likely to create a hazardous situation based on multi-camera context. Second, they propose a two-stage motion generation process that initially models a direct collision and then transforms it into a plausible near-miss or evasion trajectory, preserving the scenario's risk while ensuring it can be faithfully rendered by the video model. Third, a diffusion model is used to convert these trajectories into the final multi-view video output.

**Strengths:**

1. The designed safety-critical video generation pipeline is intuitive and important for training robust E2E model.

2. The proposed two-stage evasion trajectory generator provides diverse collision scenes.

3. The video results look consistent and dynamically plausible.

**Weaknesses:**

1. SafeMVDrive works as a data engine to provide more diverse driving scenarios for better training E2E AD systems. However, the experiment section lacks related experiments on how the generated data can improve E2E AD model's performance under long-tailed driving scenarios.

2. In Section 3.2, the authors claim VLM selection outperforms selection methods that rely on non-visual annotations in identifying physically feasible collisions. However, this capability is not shown in the experiment section. It would be better to have experiments showing this ability qualitatively or quantitatively.

3. The motivation for using VLM is not clear. The author mentions that VLM allow fast selection during inference. However, the proposed pipeline mainly serves as a data engine, so runtime efficiency is not a critical problem. Also, the paper didn't compare the running time between simulation-based selection and VLM inference.

**Questions:**

1. In line 452, the author states that they use automated annotation to identify all vehicles that can collide with the ego vehicle. How is automated annotation performed? If this process can be automated, why not just find all these vehicles at test time and randomly pick one from the set.

---

> ### Author Response · Authors · 2025-11-21
>
> We sincerely appreciate your valuable time and comments. We have uploaded the rebuttal version and marked all the changes in blue.
>
> **W1: Lack of Experiments on how SafeMVDrive improves E2E-AD planner’s Performance.**
>
> Thank you very much for your valuable suggestion. We conduct an experiment to verify the usefulness of the videos generated by SafeMVDrive for improving an end-to-end autonomous driving planner in terms of collision avoidance. Specifically, we split the generated data into training and validation sets with a ratio of roughly 4:1, and mix the training set with the original NuScenes training dataset to fine-tune the light weight E2E planner DiffusionDrive (2025 CVPR spotlight) [1] for 10 epochs. We then evaluate the model on both the NuScenes validation set and our generated SafeMVDrive validation set. From the results below, we can see that after training the end-to-end planner, collision-related metrics decrease significantly on the SafeMVDrive validation dataset, and also decrease on NuScenes validation dataset. This further demonstrates the effectiveness of our framework. The corresponding experimental results have been added to Section 4.5 of the main paper.
>
>
> | Evaluation Set | Model |  | Sample level CR(%) **↓** |  |  |  | Scene level CR **↓** |  |  | NC **↑** | TTC **↑** |
> | :---: | :---: | :---: | :---: | :---: | :---: | :---: | :---: | :---: | :---: | :---: | :---: |
> |  |  | 1 | 2 | 3 | averaged | 1 | 2 | 3 | averaged |  |  |
> | NuScenes Val | Base | 0.11 | 0.15 | 0.28 | 0.18 | 0.03 | 0.05 | 0.08 | 0.05 | 0.992 | 0.976 |
> |   | Finetuned | 0.06 | 0.11 | 0.21 | 0.13 | 0.02 | 0.03 | 0.06 | 0.04 | 0.993 | 0.979 |
> | SafeMVdrive Val | Base | 6.63 | 17.6 | 25.68 | 16.64 | 0.81 | 2.16 | 3.15 | 2.04 | 0.743 | 0.684 |
> |  | Finetuned | 3.06 | 7.4 | 12.59 | 7.68 | 0.37 | 0.91 | 1.54 | 0.94 | 0.822 | 0.763 |
>
>
> [1] Liao, B., Chen, S., Yin, H., Jiang, B., Wang, C., Yan, S., ... & Wang, X. (2025). Diffusiondrive: Truncated diffusion model for end-to-end autonomous driving. In Proceedings of the Computer Vision and Pattern Recognition Conference (pp. 12037-12047).
>
> **W2: Effectiveness of the VLM-based adversarial vehicle selector.**
>
> In the evaluation presented in Section 4.3, we have compared our VLM-based selector with rule-based methods **quantitatively**. The accuracy and recall reported in Table 2 for identifying safety-critical vehicles jointly reflect the ability to select physically feasible and safety-critical targets. The results show that incorporating visual information through the VLM provides benefits and outperforms previous rule-based selection strategies.
>
> In addition, we include **qualitative** examples demonstrating that the VLM is capable of making physically feasible selections. These examples are provided in Appendix A.6. When obvious obstacles are present, rule-based methods may incorrectly select blocked vehicles, whereas the VLM successfully avoids these vehicles and selects the correct one.
>
> **W3: Motivation for using VLM and runtime comparison.**
>
> The use of VLM is to leverage its spatial perception capability to select physically feasible vehicles as detailed in the above answer and to significantly speed up the selection process compared with exhaustive simulation. Specifically, since trajectory simulation incurs a non-trivial computational cost—approximately 2 minutes per simulation—simulating every vehicle in a scene to identify safety-critical ones is time-consuming, typically averaging around **28** minutes and exceeding **98** minutes in scenes with a large number of vehicles. Even when considered purely as part of a data engine, such simulation time constitutes a significant overhead. In contrast, once trained, the VLM requires only about **22** seconds per scene to accurately identify safety-critical adversarial vehicles, yielding roughly an **80×** speed-up. Moreover, according to the scale laws of autonomous driving [1], training a commercial planner usually requires massive datasets, often amounting to tens of thousands of hours of driving data. At such a scale, the speed of data generation becomes crucial, as it can directly impact the overall time-to-market. Finally, thank you for your suggestion. We have added the running-time comparison to Appendix A.7.
>
> [1]: Naumann, Alexander, et al. "Data Scaling Laws for End-to-End Autonomous Driving." *Proceedings of the Computer Vision and Pattern Recognition Conference*. 2025.

---

> ### Author Response · Authors · 2025-11-21
>
> **Q1: How to automate annotation? Why not reuse the automated method instead of relying on the VLM?**
>
> We exhaustively traverse all vehicles in the initial scene (excluding those that are too distant) as potential adversarial vehicles. For each candidate, we employ the collision simulation phase of our two-stage simulator to determine feasibility: if a candidate vehicle collides with the ego vehicle before colliding with any other vehicles or entering a non-drivable area, we regard it as a valid adversarial vehicle and include it as part of the ground truth. As discussed above in **W3**, using a VLM to perform the selection at test time can **greatly accelerate** the identification of safety-critical vehicles relative to exhaustively traversing simulations.

---

### Author Response · Authors · 2025-12-04
**Summary of non-feedback Rebuttal (Part 2/2): Reviewer GrpU (Rating: 6, Confidence: 4)**

Reviewer **GrpU** praises the importance of generating safety-critical multi-view videos but raises several concerns. Below is a summary of how we address them:

> **Concern 1 (Closed-loop simulation and divergence):**
>

The reviewer raises a concern about our work not being closed-loop and that our videos do not fully reflect the planner’s actual behavior.

- **Solution**: We explain that this divergence is an inherent factor of open-loop evaluation, which is a standard and efficient methodology in E2E planner evaluation. Closed-loop evaluation with video generators remains challenging due to error accumulation in video generation, and it diverges from our goal of generating high-quality safety-critical videos.
- **Conclusion**: Open-loop evaluation is valid for stress-testing planners, and closed-loop simulation is not within the scope of our work and represents a separate direction.

> **Concern 2 (Fairness of Table 2 and uninteresting results):**
>

The reviewer questions the fairness of the comparison in Table 2 and notes that the collision examples are limited to rear-end collisions.

- **Solution**: We clarify the fairness of the comparison and have improved the relevant content in Section 4.3 to make it clearer. Additionally, we explain that our generated scenarios include more complex behaviors, not just simple rear-end collisions (such as examples 1 and 6 in [https://iclr-1.github.io/SMD/#Video Gallery](https://iclr-1.github.io/SMD/#Video%20Gallery)). More importantly, rather than focusing on being "interesting", we emphasize that the primary value of these scenarios lies in their usefulness for autonomous driving, particularly for stress-testing and improving end-to-end planners.
- **Conclusion**: The comparison in Table 2 is fair, and the generated videos include more than just rear-end collisions. More importantly, they are valuable for improving autonomous driving applications, beyond simply being interesting.

> **Concern 3 (3-Second horizon in evaluation):**
>

The reviewer asks why the evaluation is limited to a 3-second horizon.

- **Solution**: We justify the 3-second horizon as part of the standard open-loop evaluation protocol, which is commonly used in the NuScenes open-loop evaluation for short-term decision-making.
- **Conclusion**: The 3-second horizon is a standard duration in the open-loop evaluation method for NuScenes.

> **Concern 4 (Auto-labeling and simulator performance):**
>

The reviewer raises a minor concerns about the potential limitations of the auto-labeling process due to the CTG simulation framework.

- **Solution**: We explain that CTG is a widely adopted simulation framework that provides sufficiently accurate trajectories for our labeling pipeline.
- **Conclusion**: The current auto-labeling process is sufficient, and we plan to explore improvements in future work.

> **Concern 5 (Closed-Loop simulation handling):**
>

The reviewer inquires whether the framework can handle closed-loop simulation.

- **Solution**: We clarify that our framework does not currently support closed-loop simulation due to the challenges of auto-regressive video generation, which often leads to error accumulation. Closed-loop simulation remains a challenging problem in video generation.
- **Conclusion**: Closed-loop simulation is a future goal, but it is currently not feasible due to the complexity of maintaining consistent video trajectories.

> **Concern 6 (Handling multi-view visual inputs with VLM):**
>

The reviewer asks how well the VLM handles multi-view visual inputs, especially side-view vehicles.

- **Solution**: We explain the scope of our VLM's capabilities, noting that our current work focuses on ensuring that the VLM can select safety-critical vehicles that lead to physically feasible collisions.
- **Conclusion**: Our VLM demonstrates partial but not yet complete capability in handling multi-view visual inputs. Enhancing side-view capabilities is an important goal for our future research.

> **Concern 7 (Inference speed):**
>

The reviewer asks for the inference speed of the proposed pipeline.

- **Solution**: We report the inference time (see **Q3** of GrpU) and explain that this is comparable to existing methods.
- **Conclusion**: The inference speed is currently comparable to existing methods, and we aim to improve it in future.

In summary, we have addressed all concerns raised by Reviewer **GrpU**.

---

### Author Response · Authors · 2025-12-04
**Summary of non-feedback Rebuttal (Part 1/2): Reviewer izyK (Rating: 4, Confidence: 3）**

Reviewer izyK praises the quality of the generated videos and the importance of our framework for generating safety-critical multi-view videos, but raised several concerns. Below is a summary of how we address them:

> **Concern 1 (Improving for E2E planners):**
>

Request experiments on how the generated data can improve E2E  AD planner performance under long-tailed driving scenarios.

- Solution: We conducted an experiment to verify the usefulness of the videos generated by SafeMVDrive for improving an E2E AD planner in terms of collision avoidance (see **W1** of izyK).
- Conclusion: After training the E2E planner, collision-related metrics significantly decreased on both the SafeMVDrive and NuScenes validation datasets. This additional experiment demonstrates the **validity and practical value** of our framework.

> **Concern 2 (Effectiveness of VLM selection):**
>

Request qualitative or quantitative evidence that the VLM-based adversarial vehicle selector outperforms rule-based selection methods.

- Solution: We provided both qualitative and quantitative experiments to support our claims. Quatitative reults can be found in **Table 2**, and a qualitative examples has been added in **Appendix A.6**.
- Conclusion：Quantitative results show that VLM outperforms rule-based selection methods.  Qualitative results show that, when obvious obstacles are present, rule-based methods may incorrectly select blocked vehicles, while VLM successfully avoids these vehicles and selects the correct one.

> **Concern 3 (Motivation for using VLM and runtime comparison):**
>

 The motivation for using VLM is unclear, and require runtime comparison between VLM inference and simulation-based selection.

- Solution：We have provided a detailed explanation of how VLM leverages spatial perception to select physically feasible vehicles, and we also added a runtime comparison (see **W3** of izyK).
- Conclusion:  The use of VLM is to leverage its spatial perception capability to select physically feasible vehicles and significantly accelerates the process, offering an **80x speedup** compared to exhaustive simulation.

> **Concern 4（Automated annotation and test-time reuse）：**
>

Require a detailed explanation of how automated annotation is performed and why it is not reused at test time.

- Solution：We explained how automated annotation is done in detail (see **Q1** of izyK) and clarified the benefits of using VLM over exhaustive simulation for efficient vehicle selection.
- Conclusion: Automated annotation improves **80x** efficiency over exhaustive search at test time, ensuring greater speed.

In summary, we have addressed the concerns raised by Reviewer **izyK**.

---

### Author Response · Authors · 2025-12-04
**Summary of Responses to Reviewer Feedback and Key Updates for Area Chair**

Dear Area Chair,

We sincerely appreciate your time and effort in reviewing our paper. To support your assessment, we provide a concise summary of our rebuttal and the corresponding reviewer interactions.

Across the reviews, all reviewers acknowledged the importance of our framework for generating safety-critical multi-view videos. Multiple reviewers highlighted the paper’s **novelty** (ffnN, GrpU), the **high quality and realism** of our generated videos (c85w, izyK), and the **clarity and organization** of our presentation (ffnN, c85w). These points indicate a strong baseline level of enthusiasm for the contribution.

During the rebuttal phase, both **Reviewer ffnN (rating 8)** and **Reviewer c85w (rating 6)** engaged actively with our responses and **explicitly expressed satisfaction** with the clarifications and additional experiments we provided. While the other two reviewers did not respond before the system reset, we addressed each of their concerns with detailed comparisons, new empirical evidence, and expanded methodological justification. Based on the added results, we believe their concerns have been fully resolved.

For your convenience, we summarize the reviewer ratings and feedback in the table below:

| Reviewer | Rating | Confidence | Feedback |
| --- | --- | --- | --- |
| **ffnN** | 8 | 4 | Positive |
| **c85w** | 6 | 4 | Positive |
| **GrpU** | 6 | 4 | No Feedback |
| **izyK** | 4 | 3 | No Feedback |

Below we summarize the key rebuttal contents that directly address reviewer concerns:

> **1. Safety-critical data improves End-to-end planning performance**
>

**Original Concern:** Reviewers izyK, ffnN, and c85w raise concerns or questions about whether the generated safety-critical data can improve the performance of the end-to-end planner.

**Our Response:** We conduct an experiment to verify the usefulness of the videos generated by SafeMVDrive for improving an end-to-end autonomous driving planner in terms of collision avoidance. Specifically, we split the generated data into training and validation sets, mixing the training set with the original NuScenes dataset to fine-tune the end-to-end planner, DiffusionDrive. We evaluate the model on both the NuScenes validation set and our generated SafeMVDrive validation set. **Our results show that collision-related metrics of the planner decrease significantly on the SafeMVDrive validation dataset and also decrease on the NuScenes validation dataset**.

This additional experiment further demonstrates the **validity and practical value** of our framework, as recognized by Reviewer c85w and Reviewer ffnN.

> **2. Evaluate a broader range of planners**
>

**Original Concern:** Reviewer c85w recommended that we should evaluate a wider range of planning methods to verify the consistency of results.

**Our Response:** We have now added two additional end-to-end planners (SparseDriveand DiffusionDrive) to evaluate the videos generated by our framework. **The experimental results are consistent: the SafeMVDrive-generated videos pose significant challenges to the planners** compared to those generated with the original trajectories, leading to a substantial improvement in collision-related metrics.

Our results demonstrate that the videos generated by our framework exhibit **strong safety-critical characteristics, posing a challenge to a wide range of end-to-end planners.**

> **3. Effectiveness and role of VLM-based safety-critical vehicle selection**
>

**Original concern:** Reviewers izyK and c85w raised concerns about the effectiveness and role of our VLM-based safety-critical vehicle selection.

**Our response:** We have provided a detailed explanation and added both qualitative and quantitative experiments to support our claims. The effectiveness of VLM is demonstrated by the following two points:

1.**By leveraging its spatial perception capability, VLM selects physically feasible vehicles.** From a qualitative perspective, our VLM-based safety-critical vehicle selection outperforms traditional rule-based methods, achieving a higher F1-score. Additionally, we include qualitative examples: When obvious obstacles are present, rule-based methods may incorrectly select blocked vehicles, while VLM successfully avoids these vehicles and selects the correct one.

2.**VLM significantly speeds up the selection process compared to exhaustive simulation.** We have added a quantitative comparison, showing that our method **accelerates the process by 80** times compared to exhaustive search.

**In summary,** reviewers recognized our paper’s novelty, importance, and high-quality results. Two reviewers explicitly confirmed their concerns were resolved after rebuttal. We believe the strengthened evidence demonstrates that our framework makes a meaningful and impactful contribution to safety-critical multi-view video generation and autonomous driving research.

Thank you for your consideration.

**Sincerely,
The Authors**

---

### Note · Authors · 2026-01-26

I have read and agree with the venue's withdrawal policy on behalf of myself and my co-authors.

---

### Meta-Review · Area_Chair_vH8a · 2026-01-06

**Summary:**

Major weaknesses and issues raised by reviewers include
1) the effect of generated safety-critical synthetic data in improving the model performance
2) VLM-based safety-critical vehicle selection
3) Evaluate more planners
4) Open-loop but not closed-loop setting (realistic setup)
5) Fairness of comparison
6) Diversity of safety-critical cases
7) Efficiency

**Reviewer Concerns:**

Still outstanding issues include (numbering aligned with above)

4) This work does not support closed-loop yet.

6) While the authors provided extra examples, It is still unclear with the distribution of safety-critical cases. More details are needed for issues like:

> a) the performance analysis of model such as why the gain varies significantly between real (NuScenes Val) and synthetic (SafeMVDrive Val) data?

> b) Why the gain in NC and TTC for NuScenes is almost neglectable (level of 0.001/0.003)?

> c) Is the fine-tuned model leaned more with synthetic data?

**Reviewer Scores:**

Three reviewers are generally positive even in the initial review round. The decision would be largely up to the one giving the lowest score (4) with focus on evaluation. In this aspect, AC also noted several additional issues:

> only safety-critical related metrics are reported and compared, but no other metrics not related to collision are evaluated. This will raise questions like how fining tuning on such synthetic data will affect the performance of latter metrics, negative or positive or little/none? Obviously this point is totally missing but is necessary to be included to make a comprehensive view for evaluation.

> Regarding the way of model training, the proposed method needs an extra fine-tuning stage after the base training. One obvious question is that what if one lets the base training use the same amount of training resources such as GPU hours, how the model will perform? This is missing but useful to disentangle out the effect of training cost from the effect of training data used.

While most of the review comments are addressed, AC still consider this work is not ready to publish for the above shortcomings.

---

### Decision · Program_Chairs · 2026-01-26

Reject